# Rare earth element resources on Fuerteventura, Canary Islands, Spain: a geochemical and mineralogical approach

Marc Campeny[1], Inmaculada Menéndez[2], Luis Quevedo[2,3], Jorge Yepes[2], Ramón Casillas[4], Agustina Ahijado[4], Jorge Méndez-Ramos[3], José Mangas[2]

[1] Departament de Mineralogia, Museu de Ciències Naturals de Barcelona, Passeig Picasso s/n, 08003 Barcelona, Spain
[2] Instituto de Oceanografía y Cambio Global, IOCAG, Universidad de Las Palmas de Gran Canaria, 35017 Las Palmas de Gran Canaria, Spain
[3] Instituto de Materiales y Nanotecnología, Departamento de Física, Universidad de La Laguna, apartado correos 456, 38200 La Laguna, Tenerife, Spain
[4] Departamento de Biología Animal, Edafología y Geología, Universidad de La Laguna, apartado correos 456, 38200 La Laguna, Tenerife, Spain.

*Correspondence to*: Marc Campeny (mcampenyc@bcn.cat)

**Abstract.** Rare earth elements (REEs) play a pivotal role in the ongoing energy and mobility transition challenges. Given their critical importance, governments worldwide and especially from the European Union, are actively promoting the exploration of REE resources. In this context, alkaline magmatic rocks (including trachytes, phonolites, syenites, melteigites and ijolites), carbonatites and their associated weathering products were subjected to a preliminary evaluation as potential targets for REE exploration on Fuerteventura Island (Canary Archipelago, Spain) based on mineralogical and geochemical studies. These lithologies show significant REE concentrations. However, only carbonatites exhibit the potential to host economically viable REE mineral deposits. REE concentrations in carbonatites of about 10,300 ppm REY (REEs plus yttrium) have been detected, comparable to other locations hosting significant deposits of these critical elements worldwide. Conversely, alkaline magmatic rocks and the resulting weathering products display limited REE contents. Notably, REEs in carbonatites are associated with primary accessory phases such as REE-bearing pyrochlore and britholite, and secondary monazite. The carbonatites of Fuerteventura hold promise as prospective REE deposits within a non-conventional geological setting (oceanic island). However, due to intricate structural attributes, the irregular distribution of these mineralizations and possible land use and environmental constraints, additional future detailed investigations are imperative to ascertain their viability as substantial REE resources.

**Keywords.** Rare earth elements, Carbonatites, Fuerteventura, Canary Islands, Weathering

## 1 Introduction

The implementation of actions to mitigate climate change is one of the major global challenges facing society today. In this regard, the European Commission (EC) has adopted a series of economic and technological proposals aimed at reducing greenhouse gas emissions and transforming Europe into the first climate-neutral continent by 2050. These guiding principles, encompassed in the European Green Deal (EGD) (European Commission, 2019), are directly linked to the establishment of an energy and mobility transition based on green technologies that will replace the current fossil fuel-based model. However, in order to achieve the ambitious targets, other finite resources will be essential: metals. Some of them, commonly referred to as *green metals*, will play a crucial role in a successful energy and mobility transition and, consequently, in achieving the neutrality goals outlined by the EGD (Graedel et al., 2015; Jyothi et al., 2020). Among the green metals is the group of rare earth elements (REEs) that are critical for a wide range of high-tech applications such as wind turbines, electric vehicles, rechargeable batteries, energy-efficient lighting, optical telecommunications, photovoltaic cells, solar energy harvesting or artificial photosynthesis for "green hydrogen" ($H_2$) generation (Méndez-Ramos et al., 2013; Acosta-Mora et al., 2018; Wondraczeck et al, 2015). Such elements are also commonly known as the *vitamins of modern industry* (Alonso et al., 2012; Chakhmouradian and Wall, 2012; Massari and Ruberti, 2013; Charalampides et al., 2015; Weng et al., 2015; Balaram, 2019).

According to the International Union of Pure and Applied Chemistry (IUPAC), REEs comprise a group of 17 chemical elements: scandium (Sc), yttrium (Y) and the 15 members of the lanthanide series (Connelly et al., 2005). The term "rare" is confusing because, even though REEs seldom occur in pure mineral phases, their average concentration in the Earth's crust is around 125 ppm, surpassing other common industrial metals such as copper, gold or platinum (Long et al., 2010; Rudnick and Gao, 2014).

Given their pivotal role in modern industry and green technologies, as well as the projected increase in demand for REEs in the coming years, governments worldwide are actively promoting the exploration of new REE resources (Barteková and Kemp, 2016). In line with this, the EC included light rare earth elements (LREEs) and heavy rare earth elements (HREEs) in the 2023 list of critical raw materials (CRMs), acknowledging them as essential and considering HREEs as the material with the highest supply risk (European Commission, 2023a).

On March 16, 2023, the EC presented a bold initiative to the European Parliament: The European Critical Raw Materials Act. This regulation aims to establish a comprehensive framework to ensure a secure and sustainable supply of CRMs, including REEs, in the coming years (European Commission, 2023b).

The search for REEs in the geological environment has primarily centred on investigating non-conventional HREE sources such as soils and weathering products (Braun et al., 1993; Wang et al., 2010, Wang et al., 2013; Berger et al., 2014; Aiglsperger et al., 2016; Torró et al., 2017; Reinhardt et al., 2018; Borst et al., 2020), but also traditional and well-known LREE-bearing lithologies, such as carbonatites (Goodenough et al., 2016; Yang et al., 2019; Pirajno and Yu, 2022).

Carbonatites are igneous rocks formed by carbonate mantle melts and are genetically associated with a wide range of mafic, ultramafic, and alkaline silicate rocks (Yaxley et al., 2002). Although carbonates such as calcite or dolomite are their main forming minerals, a significant portion of carbonatites contain accessory phases enriched in critical metals such as REEs (Christy et al., 2021). REEs can be contained in fluorcarbonates (e.g., bastnäsite, parisite, huanghoite, synchysite), phosphates (e.g., monazite, rhabdophane), silicates (e.g., allanite), or even oxides (e.g., REE-bearing pyrochlore, cerianite). These accessory minerals make carbonatites the main current REE source, representing 86.5% of the deposits under exploitation for these elements (Liu et al., 2023). However, although carbonatites are rare rocks, predominantly found in continental rifts associated with cratons (Humphreys-Williams et al., 2021), they have exceptionally been described in other geological contexts, most notably oceanic islands associated with hotspots, such as Cape Verde (Mourão et al., 2010; De Ignacio et al., 2018;) or Fuerteventura in the Canary Islands (Mangas et al., 1996; Carnevale et al., 2021).

The petrogenesis of carbonatites is still a debated topic (Anenburg et al., 2021; Yaxley et al., 2022). Different processes have been proposed for their formation, although there is a consensus that they originate from primary fusion processes derived from a carbonated mantle (Kamenetsky et al., 2021). For the specific case of the oceanic carbonatites, this debate is even more lively. Doucelance et al. (2010) suggested a shallow origin from low-degree partial melting at the base of the oceanic lithosphere. Other authors have proposed the involvement of unmixing process linking to alkaline magma suites (Weidendorfer et al., 2016), the action of hydrothermal fluids of marine origin enriched in Ca that would have serpentinized the mantle (Park and Rye, 2013) or even the contribution of recycled marine carbonates through subduction or assimilated in shallow magma chambers (Démeny et al., 1998; Hoernle et al., 2002; Doucelance et al., 2014).

The present work concentrates on the mineralogical and geochemical study of carbonatites and associated
alkaline igneous rocks, along with their weathering products, in three different sectors in the western region
of Fuerteventura (Canary Islands, Spain; Figure 1). The primary goal of this research is to conduct an initial
evaluation of these materials, which are genetically associated with a volcanic island linked to an oceanic
intraplate magmatism. This assessment aims to enhance our understanding of REE accumulation while
appraising the potential of this peculiar geological environment as a non-conventional source of REE.

**2 Geological setting**
**2.1 The Canary Island Seamount Province**
The Canary Islands archipelago, located between 27°N and 30°N of latitude, is part of the Canary Island
Seamount Province (CISP). This volcanic region forms a band of approximately 1300 km in length and
350 km in width, running parallel to the African continental margin. Within the CISP, there are over 100
seamounts and up to 8 emerged islands: El Hierro, La Palma, La Gomera, Tenerife, Gran Canaria,
Lanzarote, Fuerteventura and Savage islands (Courtillot et al., 2003; Schmincke and Sumita, 2010; van den
Bogaard, 2013). Based on magnetic anomaly measurements and dating of both emerged and submarine
igneous materials, volcanic activity in the CISP spans more than 142 Ma, from the Early Cretaceous to the
present day (Frisch, 2012; van den Bogaard, 2013; Longpré and Felpeto, 2021).
**2.2 Fuerteventura Island**
Fuerteventura, the easternmost island of the Canarian archipelago, along with Lanzarote, forms the
emergent crest of the Eastern Canarian Volcanic Ridge, which is located approximately 100 km offshore
from the Moroccan coast (Figure 1). Fuerteventura is the oldest island in the archipelago, with its initial
stages of formation linked with submarine volcanic activity, dating to the Oligocene (~34 Ma). The first
episodes of subaerial volcanism occurred around ~23 Ma (Coello, 1992; Ancochea et al., 1996; Pérez-
Torrado et al., 2023).
Fuerteventura is characterized by the occurrence of three distinct main geological units, arranged in order
from oldest to youngest: the Fuerteventura basal complex (FBC), the Miocene subaerial volcanic units, and
the Pliocene-Quaternary volcano-sedimentary facies (Fúster et al., 1968; Le Bas et al., 1986; Muñoz et al.,
2005; Gutiérrez et al., 2006; Troll and Carracedo, 2016).

**2.2.1 The Fuerteventura basal complex**
The FBC unit mainly outcrops in the western part of the island (Figure 1). Two different groups of
lithofacies may be distinguished: (1) Early Jurassic to Late Cretaceous oceanic crust materials (Steiner et
al., 1998), constituted by mid-ocean ridge basalts and oceanic sediments; (2) Oligocene submarine and
transitional volcanic rocks associated with plutonic bodies and dyke swarms (Feraud et al., 1985; Hobson
et al., 1998; Gutiérrez et al., 2006). In this second group, a set of lithologies can be distinguished related to
an ultra-alkaline-carbonatitic magmatic pulse that occurred ~25 Ma (Le Bas, 1981; Barrera et al., 1986;
Balogh et al., 1999). Additionally, alkaline ultramafic, mafic and felsic plutonic rocks such as wehrlites,
pyroxenites, gabbros and syenites intruded the previously existing Oligocene materials, forming distinctive
ring complexes (Muñoz et al., 2005). These magmatic rocks, predominantly of Oligocene age, have been
interpreted as episodes of submarine and transitional growth in Fuerteventura (Le Bas et al., 1986; Gutiérrez
et al., 2006).
In general, outcrops related with the FBC intrusive assemblage exhibit significant variations and four
distinct morphologies and characteristic textures can be identified (Fúster et al., 1968; Barrera et al., 1986;
Le Bas et al., 1986; Fernández et al., 1997; Mangas et al., 1992, 1994, 1997; Ahijado 1999; Ancochea et
al., 2004; Ahijado et al., 2005; Muñoz et al., 2005):

(1) Basaltic, alkaline and carbonatitic dykes and veins of meter-scale, decimeter-scale, and centimeter-scale, that are randomly distributed, resulting in a chaotic arrangement (Figure 2a, b). Related to the carbonatite veins and dikes, an intense fenitization may occur.

(2) Shear zones (Fernández et al., 1997), characterized by gradual or diffuse boundaries, which display assimilation structures between different rock bodies, along with the presence of mylonites, and brecciated textures resulting from deformation (Figure 2c).

(3) Pegmatitic textures developed within certain rock bodies, often containing centimeter-sized crystals of rock-forming minerals (Figure 2d).

(4) Contact metamorphism and metasomatism, as well as skarn zones that occur in deformed or undeformed carbonatites, influenced by subsequent hydrothermal fluid circulation (Ahijado et al., 2005; Casillas et al., 2008, 2011).

In addition, during Miocene magmatic pulses, alkaline plutons were formed in the central-western part of Fuerteventura Island north of the locality of Pájara (sector 2, Figure 1). These intrusions constitute typical ring complexes of alkaline magmatic rocks, including nepheline syenites, syenites, and trachytes (Muñoz, 1969). They are regarded as the most recent rocks in the FBC (Figure 1) and have been dated using the K-Ar method, yielding an approximate age of $20.6 \pm 1.7$ Ma (Le Bas et al., 1986; Holloway and Bussy, 2008).

**2.2.2 Miocene subaerial volcanic unit**

During the Miocene, Fuerteventura witnessed the formation of up to three volcanic edifices (Figure 1; Coello et al., 1992; Ancochea et al., 1996). The northern volcanic structure, referred to as the Tetir edifice, experienced two volcanic construction phases between 22 and 12.8 Ma (Balcells et al., 1994). These episodes involved the eruption of basalts, picritic basalts, oceanic basalts, trachybasalts and trachytes. In the central part of the island, the Gran Tarajal edifice developed three different construction phases spanning from 22.5 to 14.5 Ma (Balcells et al., 1994). On the Jandía Peninsula, in the southern part of the island, a volcanic edifice comprising both basaltic and trachybasaltic materials emerged. It formed three successive construction episodes occurring between 20.7 and 14.2 Ma ago (Balcells et al., 1994). Based on their mineralogical and petrological features, the lithologies comprising this unit have not been considered as potentially containing significant concentrations of REEs. Therefore, they have not been included in the evaluation conducted in the present study.

**2.2.3 Pliocene and Quaternary volcano-sedimentary facies**

After the subaerial volcanic activity during the Miocene, a period of volcanic quiescence ensued, leading to the erosion of the previously formed volcanic edifices. Subsequently, during the Pliocene (between 5.3 and 2.6 Ma), a phase of magmatic rejuvenation began, characterized by scattered Strombolian eruptions (Figure 1). Concurrently, various sedimentary formations emerged across the entire island, including littoral and shallow-water marine deposits, as well as aeolian, colluvial, and alluvial subaerial sediments and paleosols from the Pliocene to the Quaternary (Fúster et al., 1968; Zazo et al., 2002; Ancochea et al., 2004). The soils on Fuerteventura are predominantly classified as eutric cambisols and lithosols-vitric andosols, according to the FAO/UNESCO (1970) nomenclature. However, the current arid and deforested conditions have led to extensive erosion of the weathered rock profiles present in different areas of the island. Edaphic calcretes are abundant in Fuerteventura (Alonso-Zarza and Silva, 2002; Huerta et al., 2015), with their primary source of calcium believed to be the Pliocene paleodunes formed by calcarenites, rather than the parent igneous rock itself (Chiquet et al., 1999; Huerta et al., 2015; Alonso-Zarza et al., 2020). Interestingly, the aeolian dust deposits predominantly originate from the Sahara Desert (Goudie and Middleton, 2001; Menéndez et al., 2007; Scheuvens et al., 2013).

**3 Materials and Methods**

**3.1 Sampling**

Alkaline magmatic rocks and especially carbonatites are considered potential targets for the exploration of rare earth elements (Goodenough et al., 2016; Balaram et al., 2019; Anenburg et al., 2021; Beland and Jones, 2021). In Fuerteventura, these types of lithologies are found in two distinct geological areas: the Oligocene (sectors 1 and 3; Figure 1) and the Miocene lithologies related with the FBC (sector 2, Figure 1).

Considering that weathering profiles may concentrate REE in larger quantities than primary bedrocks (Bao and Zhao, 2008; Menéndez et al., 2019, Braga and Biondi, 2023; Chandler et al., 2024), these lithological formations were included in the present evaluation study and sampling was conducted on a selection of six

different profiles: (1) Agua Salada ravine (sector 1) and (2) Aulagar ravine (sector 3), developed on carbonatites, (3) the FV-30 road, (4) Las Peñitas quarry, (5) Palomares ravine and (6) the Pájara profiles, on syenite bedrock (Figure 1; Table S1).

Accordingly, a systematic sampling campaign was conducted in three different sectors of Fuerteventura, targeting alkaline and carbonatitic igneous rocks and their associated weathering products. The specific locations of these predetermined sectors are outlined in Figure 1. As a result, a set of 29 representative samples of potentially REE-enriched magmatic rocks, along with 21 samples of associated weathering products, were collected for further analysis and evaluation (Table S1). For the weathering products, we conducted six sampling profiles (labelled A to F; Figure 1) at various suitable points to compare the mineralogical and geochemical changes resulting from weathering of the primary magmatic rocks.

## 3.2 Petrographic and mineralogical studies

Selected samples of magmatic rocks were prepared in thin sections for textural and mineralogical analysis at the Laboratory of Geological and Paleontological Preparation of the Natural Sciences Museum of Barcelona (LPGiP-MCNB; Barcelona, Spain). A representative subset of these samples was also examined using a JEOL JSM-7100 field emission scanning electron microscope (FE-SEM) at the Scientific and Technological Centers of the Universitat de Barcelona (CCiTUB). The FE-SEM system is equipped with an INCA Pentaflex EDS (energy dispersive spectroscopy) detector (Oxford Instruments, England), which allowed for the acquisition of semi-quantitative analyses of mineral phases. The general operating conditions for the FE-SEM were a 15-20 kV accelerating voltage and a 5 nA beam current.

To achieve accurate and precise mineralogical identification and characterization of the weathering magmatic rocks and calcretes, X-ray powder diffraction (XRPD) measurements were performed using a PANalytical Empyrean powder diffractometer equipped with a PIXcel1D Medipix 3 detector at the Integrated XRD Service of the General Research Support Service of La Laguna University, Spain. The diffractometer employed incident Cu $K_\alpha$ radiation at 45 kV and 40 mA, along with an RTMS (real-time multiple strip) PIXcel1D detector with an amplitude of 3.3473° 2θ. The diffraction patterns were obtained by scanning random powders in the 2θ range from 5° to 80°. Data sets were generated using a scan time of 57 seconds and a step size of 0.0263° (2θ), with a 1/16° divergence slit. Mineral identification and semi-quantitative results were obtained using the PANalytical's HighScore Plus search-match software (v. 4.5) with a PDF+ database.

**3.3 Geochemical analyses**

The major elements composition of carbonates from carbonatites was studied using an electron probe microanalyzer (EPMA) system. The EPMA analyses were conducted on a JEOL JXA-8230 electron microprobe, equipped with five wavelength-dispersive spectrometers and a silicon-drift detector EDS, located at the CCiTUB. The spot mode was employed for the analyses and the electron column was set to an accelerating voltage of 15 kV and a beam current of 10 nA. Standard counting times of 10 seconds were used, along with a focused beam, to achieve the highest possible lateral resolution. The analytical standards employed during the analysis process were: celestine (PETJ, Sr $K_\alpha$) wollastonite (PETL, Ca $K_\alpha$), periclase (TAPH, Mg $K_\alpha$), hematite (LiFH, Fe $K_\alpha$), rhodonite (LiFH, Mn $K_\alpha$) and albite (TAPH, Na $K_\alpha$).

Bulk-rock geochemical data of major and trace element composition were obtained by X-ray fluorescence (XRF) and inductive coupled plasma (ICP)-emission spectrometry. The samples were prepared by lithium metaborate/tetraborate fusion and nitric acid digestion at the ACTLABS Activation Laboratories Ltd. (Ancaster, Canada).

**4 Results**

**4.1 Petrography and mineralogy**

**4.1.1 Alkaline magmatic rocks and carbonatites**

The primary lithologies under study, consist of Oligocene (~25 Ma) alkaline igneous and carbonatitic rocks, as well as Miocene alkaline lithologies (K–Ar age of 20.6±1.7 Ma; Le Bas et al., 1986), that form part of the FBC. Their outcrops extend across kilometer-scale areas but exhibit high heterogeneity at a detailed level due to the occurrence of numerous small intrusions, ranging in size from metric to decimetric dimensions (Figs. 2a, b).

At a mineralogical level, separation of the different types of alkaline rocks found in the FBC is complex because these lithologies are intimately associated and infiltrate diffusely, leading to the formation of hybrid intrusions. The materials with the most mafic composition correspond to pyroxenites and melteigites, and their formation is associated with the earliest magmatic fractions. However, these are commonly spatially associated with more differentiated rocks, mainly ijolites, nepheline syenites, and syenites. All these lithologies have a relatively simple mineralogy, characterized by varying proportions of nepheline (10-30% modal) and potassium feldspar (50-80% modal), associated with aegirine-augite and biotite (10-30%

modal). A set of accessory minerals with varying proportions (always less than 5% modal) also occur,
including ilmenite, titanite, zircon, and fluorapatite.
At a textural level, the alkaline series lithologies of the FBC present granular textures with millimeter-sized
euhedral grains. However, in some of the intrusions in sectors 2 and 3, pegmatitic syenites-ijolites were
detected with centimeter-sized grains characterized by the presence of large aegirine-augite crystals.
Some of the intrusions described in the three sectors show aphanitic textures caused by faster cooling,
resulting in rocks with similar mineralogy but extrusive-type textural characteristics. Therefore, due to their
textural features, some dikes and apophyses, although mineralogically equivalent, should be classified as
trachytes and phonolites.
Carbonatitic intrusions commonly co-occur with the alkaline rocks, sharing similar morphology, textures,
and spatial distribution within the outcrops (Figure 2e). Furthermore, alkaline and/or carbonatitic intrusions
can be occasionally associated with mafic intrusions, primarily pyroxenites and alkaline gabbros. In
addition, a subsequent set of mafic dikes with basaltic composition overlaps the previous intrusive bodies
(Figs. 2a, b).
All carbonatites described in different outcrops from sectors 1 and 3 are predominantly composed of calcite
(95% modal) and can thus be classified as calciocarbonatites (Le Maitre, 2005). None of the studied samples
shows the occurrence of ferromagnesian carbonates such as ankerite, dolomite, and/or siderite, as well as
REE carbonates. Texturally, calcite occurs as euhedral crystals ranging in size from millimetres to
centimetres, often recrystallized and exhibiting polysynthetic twinning. In some cases, a secondary micritic
calcite matrix is present, filling interstitial spaces and fractures.
The major element composition of calcite is relatively consistent across all the carbonatite samples.
Notably, there are significant contents of SrO, with values of up to 5.43 wt%, while REEs are absent from
the carbonate composition (Table S2).
The accessory mineralogy (~5% modal) comprises disseminated phases within the calcium carbonate.
Among them, the occurrence of minerals from the spinel group, including magnetite (Figure 3a), and
primarily jacobsite, occurring as subhedral crystals of up to 50 μm (Figure 3b). Another characteristic
mineral is perovskite, occurring as subhedral crystals of up to 100 μm. These grains are remarkable for
their significant Nb contents, as described in other carbonatitic localities worldwide (Torró et al., 2012).
Britholite also occurs as subhedral crystals of up to 100 μm (Figure 3b). This primary britholite contains
significant LREE content (Figure 4), and its alteration leads to the formation of secondary REE-enriched

phosphates, mainly monazite-Nd (Figure 3c), which also contains substantial amounts of La and Ce (Figure 4). REEs, in addition to occurring in primary britholite and secondary monazite, were also detected in tiny pyrochlore grains, heterogeneously disseminated in the calcite groundmass (Figure 3b). In some cases, pyrochlore forms euhedral crystals of up to 20 μm, also included in calcite (Figure 3d). This pyrochlore shows slight zoning towards plumbopyrochlore (Christy and Atencio, 2013), with significant enrichment in Pb observed at grain borders (Figure 3d).

Carbonatites can be affected by certain contact metamorphism, especially in sectors 1 and 3 (Figure 1) and may exhibit a slightly different mineralogy from the one described thus far. This is characterized by the occurrence of skarn-type metamorphic minerals, formed due to the interaction between carbonatites and spatially associated silica-rich rocks. Among these minerals, there are subhedral crystals of andradite, up to 30 μm in size, implanted in a matrix of secondary calcite and phlogopite, exhibiting pronounced zoning with kerimasite cores (Figure 3e). In these areas, the occurrence of REE mineralizations associated with allanite (Figure 3f) is also typical. Allanite occurs as granular aggregates associated with hydrothermal secondary sulfates, primarily baryte (Figure 3f), but occasionally celestine (Figure 3c).

This particular mineralogy, typically associated with skarn formations, emerges from the interaction between a carbonatite intrusion and surrounding silicate rocks, in contrast to the typical process. It has recently gained attention from several researchers in various carbonatite locations worldwide, who have coined the term antiskarn to describe it (Anenburg and Mavrogenes, 2018; Yaxley et al., 2022).

### 4.1.2 Weathering products

In certain areas within the three studied sectors (Figure 1), there is evidence of the development of characteristic shallow geological formations consistently associated with weathering, which affect the outlined magmatic lithologies (Figs. 5, 6). These geological products were studied through the analysis of six alteration profiles, developed on carbonatites (Agua Salada and Aulagar) and syenites (Palomares ravine, FV-30 road, Las Peñitas quarry, and Pájara) (Figure 1).

The carbonatite-calcrete sections generally consist of centimetre-scale calcrete veins injected into the bedrock, seemingly without any apparent connection to the current upward lithosol (Figure 5). In general, the development of soils or weathering products was not detected on carbonatites in any of the studied sectors of the FBC.

Weathering products developed on syenite bedrock are generally more abundant, and the corresponding
alteration profiles are better preserved than in carbonatites. The cambic B horizon displays reddish to
yellowish colorations (5YR6/6), with a thickness of up to 20-30 cm. Additionally, it is common to find BC
horizons instead of B horizons, while the C horizon is well-developed, reaching a 30-40 cm thickness at
certain levels of the profile (Figure 6). Furthermore, except for the Las Peñitas profile (E profile, Figure 1),
centimetre-scale calcrete bands (Bk; Jahn et al., 2006) were also detected in deeper layers across all the
studied profiles.
In terms of mineralogical composition, carbonatite profiles exhibit significant changes due to weathering.
In general, weathering processes lead to a reduction in calcite, the disappearance of fluorapatite, and the
formation of secondary minerals like palygorskite (Figure 7). The contribution from lateral slope movement
is also evident through the presence of residual plagioclase and clinopyroxene.
In the case of syenite weathering profiles, illite/chlorite and kaolinite are the predominant secondary
products, followed by muscovite and palygorskite (Figure 7). Other minerals such as quartz were also
detected, even in the C horizons.

**4.2 Bulk-rock and mineral geochemistry**
Chemical analysis of the major, minor and trace elements were carried out in order to evaluate the
geochemical features and the distribution of REEs, on 25 representative samples of igneous rocks from the
FBC, including trachytes, phonolites, syenites, ijolites and carbonatites (Table S3). In addition, we also
analysed 21 samples of weathering products (Table S4).
The total REY (REEs plus yttrium) content in the FBC igneous rocks exhibits widespread and significant
enrichment in comparison to the average crustal values (~125 ppm, Rudnick and Gao, 2014). Notably, the
extrusive and magmatic alkaline lithologies (trachytes and phonolites as well as syenites and ijolites) show
variable REY values ranging between about 230 and 1,400 ppm (Table S3). In contrast, the carbonatitic
rocks exhibit REY content more than ten times greater than the alkaline lithologies, with specific samples
reaching maximum values of up to about 10,300 ppm, as evidenced in sample 85a sourced from a
carbonatite outcrop in sector 1 (Table S3).
The weathered magmatic rocks, though moderately significant in REY content relative to the average
crustal values (Table S4), still exhibit slightly lower levels compared to the content observed in the
associated alkaline and carbonatitic protoliths (Table S3). A contrasting pattern emerges in the calcretes,
where REY values experience a sharp reduction, presenting virtually negligible values ranging between 20
and 72 ppm REY. These levels are significantly below the average Earth's crust values (Rudnick and Gao,
2014) and are markedly lower than those observed in both the alkaline lithologies and, particularly, the
carbonatites of the FBC.
REE normalized diagrams further underscore this distribution, portraying elevated content in the
carbonatites, followed by the alkaline rocks (Figure 8a). Meanwhile, the weathered magmatic rocks and
calcretes (Figure 8b) display significantly lower values. All studied lithologies exhibit clear negative
patterns, indicative of enrichment in LREEs relative to HREEs. Notably, carbonatites and alkaline rocks
(Figure 8a) exhibit a flattening of these negative patterns in the final segment, indicating a certain degree
of HREE enrichment.
The FCB carbonatites exhibit a depletion in some critical elements commonly associated with this lithology
such as Nb or Ta (Table S3). Negative anomalies of both Nb and Ta are clearly observed in the multi-
element diagrams of carbonatite samples (Figure 9a). However, given the presence of pyrochlore in the
carbonatites, these anomalies in Nb and Ta are likely not indicative. We interpret that the low concentrations
of these elements could be attributed to an analytical artifact that would underestimate the contents of High
Field Strength Elements (HFSE) due to the challenge of pyrochlore dissolution in the analytical digestion
protocols employed. These protocols have been primarily devised to assess the contents of REEs rather
than HFSE. Additionally, alkaline rock patterns also show a distinctive negative anomaly in Sr (Figure 9b).
As for the weathering products, their contents of other minor elements do not indicate significant
concentrations of metals or critical elements like Nb or Ta (Table S4). The multi-element diagrams for the
calcretes exhibit a negative Ta anomaly (Figure 9c), while the patterns of weathered magmatic rocks do not
reveal notable anomalies in any group of elements (Figure 9d).
A specific geochemical study of REE distribution in the six studied weathering profiles was also conducted
(Figure 10). The main objective was to evaluate the geochemical interactions between the protolith and the
related weathering lithologies, with the aim of detecting potential REE enrichments or depletions caused
by weathering processes.
In the exchange patterns of calcretes spatially associated with carbonatitic protoliths, as analyzed in the
Agua Salada and Aulagar ravine profiles (Figs. 10A, B), REE concentrations are two orders of magnitude
lower than in the carbonatite (Figure 10A), as also previously determined from the REE diagrams (Figure
8). Notably, it was found that the REE concentration is directly proportional to the distance from the
protolith (Figure 10B), and calcrete samples with the highest REE concentrations (sample 14; Figure 10)
were found in closer proximity to the primary carbonatites than more REE depleted samples (samples 15
and 18; Figure 10B). In addition, although the values of all elements are depleted in the calcrete patterns,
there is a greater depression in LREE than in HREE relative to the protolith, resulting in typically positive
patterns, except for sample 76 from the Agua Salada ravine, where a clear inverse trend is observed (Figure
10A).
In general, the diagrams in Figure 10 show that weathering products on syenites exhibit enrichment relative
to the protolith (green areas in Figs. 10C, D, E). However, calcrete samples, whether derived from
carbonatites or syenites, consistently show depletions compared to the protolith contents (reddish areas in
Figs. 10C, D, F). The diagrams corresponding to the weathering products generated on syenites exhibit
similar morphologies (Figs. 10C, D, E, F). Overall, these lithologies are characterized by enrichment in
REEs relative to the protolith as well as V-shaped patterns, featured by the presence of a negative anomaly
in Eu, which is also reported in all C and B horizons developed on syenites, except sample 61 (Figure 10C),
and is likely related to plagioclase crystallization.

**5 Discussion**
**5.1 REE evaluation of the FBC magmatic rocks**
The FBC magmatic rocks, in the three study sectors, encompass alkaline lithologies (trachytes, phonolites,
syenites, melteigites, and ijolites) as well as carbonatites. Regarding the group of alkaline rocks, the
detected REE content varies between 214 and 1,330 ppm (Table S3), significantly higher than the average
concentration determined in the Earth's crust (~125 ppm, Rudnick and Gao, 2014). However, this finding
is not surprising, and the observed values in Fuerteventura are not anomalous, as these types of lithologies
typically exhibit REE concentrations within this range (Dostal, 2017). Therefore, the measured REE
concentrations are neither significant nor sufficiently elevated to hypothetically consider these lithologies
as a potential non-conventional deposit of these critical elements in the FBC.
On the other hand, FBC carbonatites present significantly higher values in terms of REE content. In the
studied carbonatite samples from sectors 1 and 3 (carbonatites do not outcrop in sector 2), REE content
ranges between about 1300 ppm and 10,300 ppm. The latter value corresponds to the richest REE-detected
sample in the entire FBC, which is located in the Agua Salada ravine area of sector 1 (Table S3; Figure 1).
The reported REE content values in the FBC carbonatites are similar to the general average concentrations
found in other locations worldwide where carbonatites are exploited for REE extraction. This is the case,
for example, of Bayan Obo, the largest REE mine in the world in terms of reserves and production (Lai et
al., 2015; Liu et al., 2018). In this locality, high-grade carbonatites exhibit average concentrations of 2880
ppm (Wu et al., 2008; Smith et al., 2015, 2016), which are equivalent to those measured in some of the
samples from Fuerteventura. It should be noted that low-grade carbonatite ore from Bayan Obo presents
extremely high values in comparison to the FBC, with REE concentrations reaching 30,750 ppm (Chao et
al., 1997; Smith et al., 2016).
Another significant example is the Mountain Pass carbonatite in California, USA, regarded as the largest
REE mine in the American continent, intermittently in operation for REE extraction since 1954 (Olson et
al., 1954; Haxel, 2005). In this REE deposit, average value across the whole complex are around 2580 ppm
(Castor et al., 2008; Mariano and Mariano, 2012; Smith et al., 2016), also in line with REE concentrations
detected in the present study for the FBC carbonatites.
This comparative analysis can also be carried out using normalized REE values (Figure 11). In this regard,
FBC carbonatites are significantly depleted in LREE compared to those from Bayan Obo (Yang et al.,
2019) and Mountain Pass (Castor et al., 2008), although they show similar values to other REE deposits
associated with carbonatites, such as those in Ashram, Canada (Beland and Jones, 2021) and Bear Lodge,
USA (Moore et al., 2015; Smith et al., 2016; Figure 11). However, the pattern of the Fuerteventura
carbonatites exhibits a slightly less pronounced slope, indicating a higher relative content of HREE, which
are considered the materials with the highest risk of supply among all the CRMs defined by the EC
(European Commission, 2023a). In fact, in the FBC carbonatites, the normalized HREE values are
equivalent to those reported in the primary carbonatitic rocks from the deposits of Bayan Obo (China) and
Mountain Pass (USA) (Figure 11). The relative significant HREE content reported in FBC carbonatites
holds particular significance for several economic and technological reasons. The use of HREEs, such as
Yb, Er, and Tm, is of particular interest in cutting-edge photonic and nanotechnology applications.
At the mineralogical level, it was observed that, in the FBC carbonatites, the main REE-hosting minerals
are accessory phases; primarily minerals from the pyrochlore group, found as disseminated euhedral micro-
crystals implanted in primary calcite (Figs. 3b, d). Another REE-bearing mineral in the FBC carbonatites
is britholite, which exhibits significant LREE content. However, this mineral is commonly altered to

monazite (Figs. 3c, 4), interpreted as a secondary phase but also a carrier of these critical elements (Chen et al., 2017).

Another noteworthy aspect is the lack of REE fluorcarbonates like bastnäsite $REE(CO_3)F$, parisite $Ca(REE)_2(CO_3)_3F_2$, synchysite $Ca(REE)(CO_3)_2F$ or huanghoite $Ba(REE)(CO_3)_2F$. They do not occur in the FBC, as they do in other REE deposits associated with, for example, the Bayan Obo carbonatite or the Sulphide Queen carbonatite from Mountain Pass (Castor et al., 2008; Smith et al., 2015, 2016). This point is crucial for a future hypothetical evaluation of the FBC carbonatites, as the processing of oxides and phosphates for REE extraction is a much more complex and expensive treatment process than for REE-bearing carbonates (McNulty et al., 2022).

**5.2 REE evaluation of associated weathering products**

The weathering materials developed on magmatic rocks, also analysed for their REE concentrations, constitute the remnants of soils that were interpreted as developed under wetter conditions during a humid phase of the oxygen isotope stage 2, spanning from 29 to 20 thousand years BP (Huerta et al., 2016). This period aligns with the last glacial maximum, marked by heightened humidity in the Canary Islands, resulting in slope erosion and the formation of talus flatiron (Gutiérrez-Elorza et al., 2013). Over time, these materials have undergone substantial volume reduction due to human-driven deforestation and erosion, primarily before the 15[th] century (Machado-Yanes, 1996). Notably, topography plays an essential role in the distribution of these weathering profiles and influences specific physical attributes such as slope (FAO/UNESCO, 1974).

The studied weathering products developed on syenite rocks (profiles C, D, E, F; Figs. 1, 7) are classified by FAO/UNESCO (1974) as eutric cambisols, reflecting a Mediterranean climate condition. Indeed, on the African continent, which is adjacent to the Canary Islands, eutric cambisols are primarily found within the tropical subhumid zone, gradually transitioning into the semi-arid zone (FAO/UNESCO, 1974). These syenite weathering profiles exhibit better-preserved characteristics and a more significant extent compared to those studied in carbonatites (profiles A and B; Figs. 1, 7). In general, intensive weathering plays a crucial role in the formation of REE deposits, as these elements tend to be concentrated in such geological formations compared to others leached during the weathering process. This phenomenon is exemplified in several locations worldwide, where REE deposits associated with weathering products occur: for instance, Bear Lodge in the USA (Andersen et al., 2017), Chuktukon and Tomtor in Russia (Kravchenko and

Pokrovsky, 1995; Kravchenko et al., 2003; Chebotarev et al., 2017), Las Mercedes in the Dominican
Republic (Torró et al., 2017), Araxá in Brazil (Braga and Biondi, 2023), and Mount Weld in Australia
(Zhukova et al., 2021; Chandler et al., 2024), among many others. However, the weathering processes on
Fuerteventura are characterized by fluctuating climatic conditions and intense erosion in the context of a
typical Mediterranean climate, which is in turn characterized by drier conditions and a lower propensity for
intense weathering compared to tropical climates. The weathering processes on Fuerteventura do not
therefore typically lead to the development of laterites and mature weathering profiles, since these
conditions do not favor the formation and subsequent preservation of these products, particularly within
the carbonatite bedrock areas. Consequently, this constraint substantially reduces the capacity of the FBC
to potentially contain economically valuable REE concentrations within the associated weathering
products.

**5.3 Fuerteventura carbonatites as potential REE source**
Based on the mineralogical and geochemical data, it can be concluded that, among the lithologies studied
in the FBC, only the carbonatites are favorable targets for further characterization and evaluation of their
potential economic viability as an REE source. Therefore, the primary alkaline rocks, as well as the entire
suite of corresponding secondary weathering products, can be ruled out.
The geochemical data obtained from the oceanic carbonatites of Fuerteventura, exemplified in multielement
and REE diagrams (Figure 8), suggest a petrogenetic affinity with carbonatites associated with
intracontinental rift geological settings. This similarity has also been previously highlighted by other
authors such as Carnevale et al. (2021) who, based on stable isotope data ($\delta^{13}C$ and $\delta^{18}C$) and noble gases
isotopic composition (He, Ne, Ar), suggested that oceanic and continental carbonatites were comparable in
petrogenetic terms. Therefore, despite the lingering questions about the formation processes of oceanic
carbonatites, their assessment as a possible source of critical metals, especially REEs, should be fully
considered in the same way as their continental counterparts.
However, when considering a more detailed assessment of the sectors where the FBC carbonatites outcrop,
it is essential to note that the distribution of these outcrops and thus potential REE mineralization is not
straightforward. The carbonatite outcrops have a very limited surface distribution, in the order of meters
(Figure 2e), and exhibit complex structural features influenced by shear metamorphism (Figure 2c) and
overlapping episodes of intrusive activity that resulted in swarms of dikes with intricate distributions (Figs.
2a, b). Hence, these general features of the carbonatite outcrops make it imperative to validly estimate their
volume and to carry out more precise studies of their depth distribution, which likely involve drilling and
geophysical techniques. These prospective hypothetical findings would provide a deeper understanding of
the morphology and dimensions of the carbonatitic bodies, enhancing the ability to calculate resources and
reserves while refining the general metallogenic modeling.
However, it is important to highlight that any attempt to assess potential REE deposits linked to FBC
carbonatites must consider the irregular distribution of these mineralizations. In addition, it should also be
considered the existence of regulatory constraints that may stem from the allocation of land for strategic
military activities, as well as environmental considerations to safeguard natural and marine-coastal areas,
especially bearing in mind that Fuerteventura is a UNESCO biosphere reserve territory. This latter point is
particularly pertinent for a specific area within sector 3 (Figure 1). Therefore, any comprehensive analysis
of the potential of FBC carbonatites as REE sources must also factor in these potential restrictions tied to
land use regulations aimed at upholding the broader socio-economic, environmental, and societal interests
inherent to a distinctive site like the island of Fuerteventura.

**6 Conclusions**
A preliminary evaluation of rare earth element (REE) content was conducted through a mineralogical and
geochemical study of alkaline and carbonatitic igneous rocks within the Fuerteventura basal complex
(FBC), along with associated weathering products. Based on the gathered data and their corresponding
interpretations, our findings can be summarized as follows:
(i)      The concentrations of REEs present in the alkaline and carbonatitic rocks of the FBC are

significant and exceed the average values attributed to the Earth's crust.

(ii)     The weathering products developed on these magmatic rocks do not exhibit significant REE

enrichment.

(iii)    Calcified horizons (Bk, calcretes), spatially related with carbonatites, have practically

negligible concentrations of REE elements. Colluvial processes may have influenced the

lateral transport and accumulation of REEs in Pleistocene-Holocene deposits distant from the

source area.

(iv)     Among the magmatic rocks, carbonatites are the only lithology studied within the FBC with

a real potential to host REE mineral resources. The detected concentrations of REY in

carbonatites range up to about 10,300 ppm, which is a comparable concentration to other locations hosting significant deposits of these critical elements worldwide.

(v)     Within carbonatites, REEs are primarily hosted in two accessory mineral phases: (1) oxides belonging to the pyrochlore group; and (2) phosphates. In this second group, primary phases such as REE-bearing britholite can be distinguished, as well as monazite generated as a secondary product from the britholite alteration.

(vi)    Primary calcite in the Fuerteventura carbonatites is not the predominant host of REEs. It displays a highly homogeneous composition with insignificant Fe-Mg content and negligible REEs.

(vii)   The carbonatites within the FBC could be considered potential REE resources associated with a non-conventional geological setting. However, the complex structural features of the studied FBC outcrops (deformation, metamorphism, swarms of dikes from different intrusive pulses...) make it essential to conduct more detailed studies to quantify the real economic possibilities of this lithology as an REE source.

(viii)  All the studied sectors contain outcrops located in restricted areas due to environmental or military use concerns. Any further detailed evaluation of the FBC carbonatites must take into account the environmental, socio-economic, and geostrategic factors that will significantly limit the real potential extension of REE deposits, considering a hypothetical exploitation.

**Acknowledgements**

This research was funded by the "Tierras Raras" project (SD-22/25) and the "MAGEC-REEmounts" project (ProID-20211010027) of the Canarian Agency for Research, Innovation and Information Society (ACIISI by its initials in Spanish) of the Canary Islands Government. Funding support was also provided by the project "Materials for Advanced Energy Generation" (ENE2013-47826-C4-4-R), "3D Printed Advanced Materials for Energy Applications" (ENE2016-74889-C4-2-R) and "Estudio de los procesos magmáticos, tectónicos y sedimentarios involucrados en el crecimiento temprano de edificios volcánicos oceánicos en ambiente de intraplaca" (CGL2016-75062-P), all funded by the Government of Spain. The collection of samples in specific protected areas required authorization from the Fuerteventura Island Government. We appreciate the cooperation and assistance provided by the Spanish Army, especially by the soldier Liberto Yeray Puga Acosta, who facilitated our access to the Pájara CMT restricted military

area to carry out sampling. We also thank Gerard Lucena from the LPGiP-MCNB for his thorough work in
the elaboration of polished thin sections.

**Statements and Declarations**
**Data availability statement**
The authors confirm that the data supporting the findings of this study are available within the article and
its supplementary materials.

**Competing interests**
The authors declare no competing interests. The funders had no role in the design of the study, in the
collection of samples, the analyses, the interpretation of data, the writing of the manuscript nor the decision
to publish these results.
**Author contributions**
Conceptualization: MC, IM, LQ, JY, JM; fieldwork and sampling: MC, IM, LQ, JY, RC, AA, JM;
methodology: MC, IM, JY, JM; validation of results: MC, IM, LQ, JY, RC, JMR, JM; data curation: MC,
IM, JM; writing-original draft preparation: MC, IM, JY, JM; writing-review editing: MC, IM, LQ, JY, RC,
JMR, JM; supervision: IM, JY, JM; project administration: JMR, JM; funding acquisition: IM, JY, RC,
JMR, JM.

**Additional information**
Supplementary tables are available in the online version at https: XXXXX

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

**Figure 1**

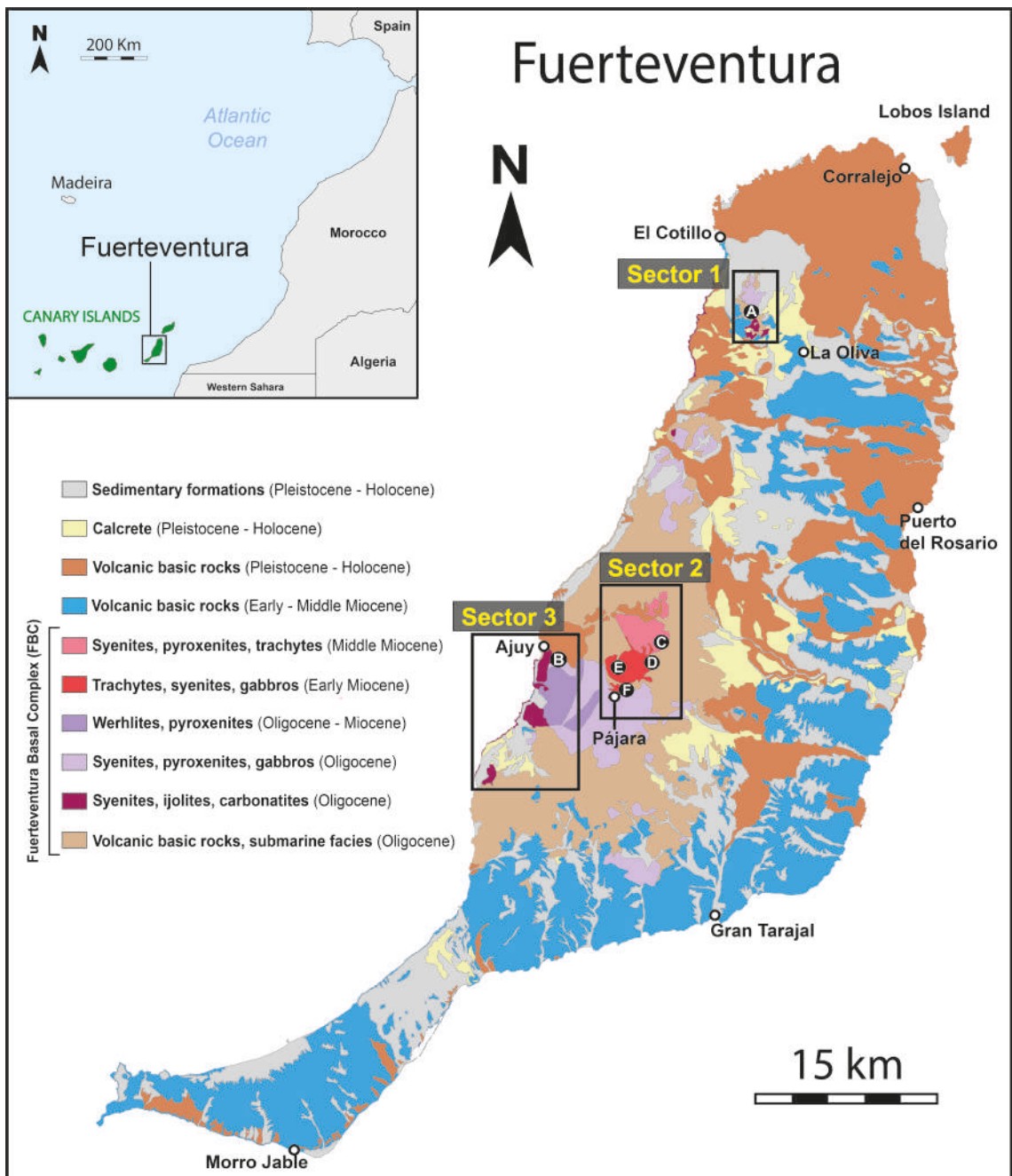

**Figure 1**: Simplified geological map of Fuerteventura Island (modified from Balcells et al., 2006) showing the location of the three study sectors for the assessment of REE content in the FBC. Additionally, the studied weathering profiles are also indicated, as: (A) Agua Salada ravine; (B) Aulagar ravine; (C) Palomares ravine; (D) FV-30 road; (E) Las Peñitas quarry; (F) Pájara.

**Figure 2**

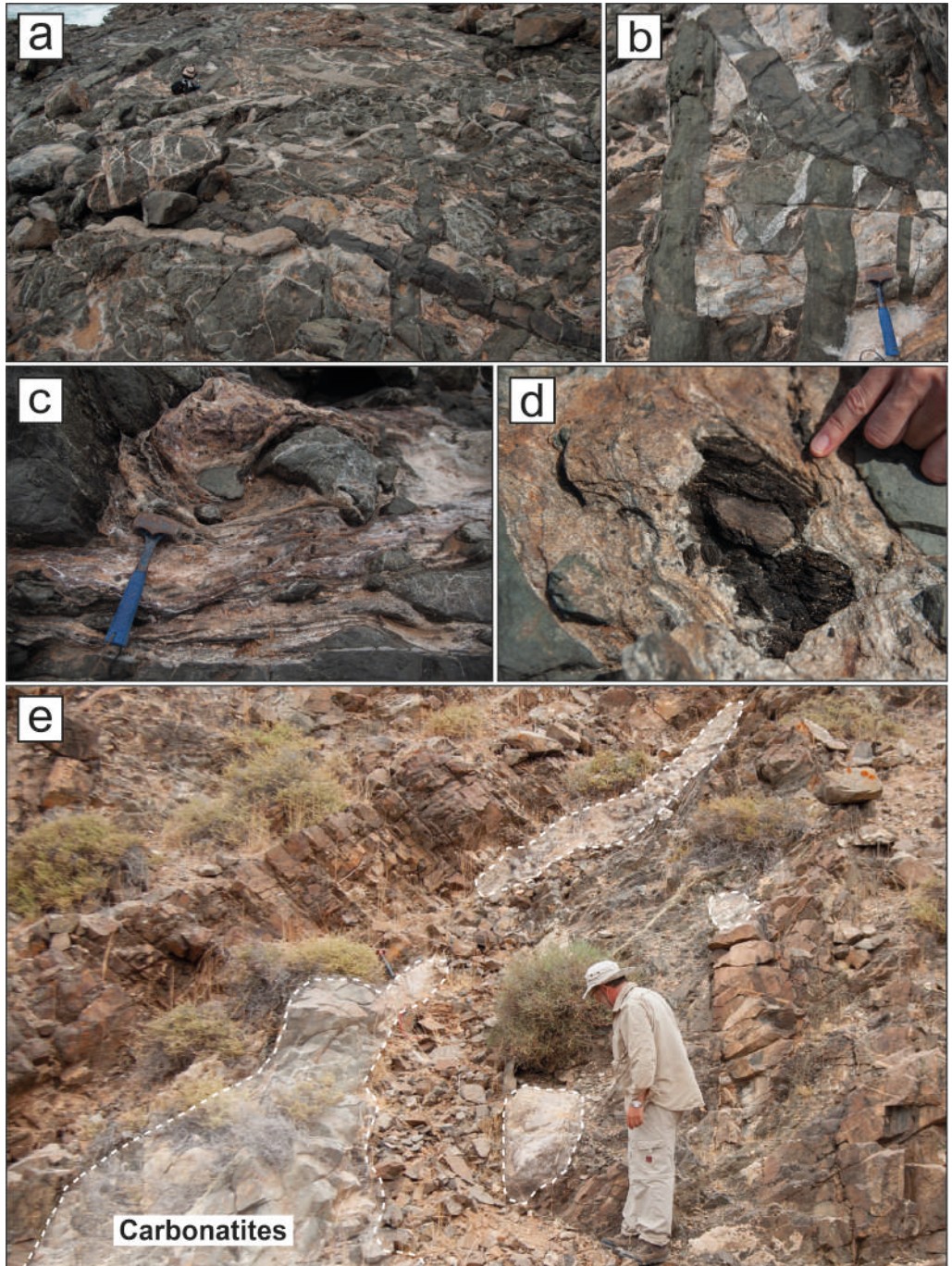

**Figure 2: (a)**, **(b)** Images showing typical outcrops of the FBC in the southern area of Ajuy (sector 3; Figure 1). The images highlight characteristic swarms of alkaline and carbonatitic intrusions (whitish) intersected by later-intruded basaltic dikes (black colour). **(c)** Detailed view of a carbonatitic dike located in a shear zone of sector 3, exhibiting distinct linear sigmoidal structures resulting from deformation. **(d)** Detailed view of centimetre-sized phlogopite crystals within a carbonatitic dike outcropping in sector 3, displaying a typical pegmatitic texture. **(e)** Overview of an outcrop of metric-scale carbonatitic dikes in the sector 1 area.

**Figure 3**

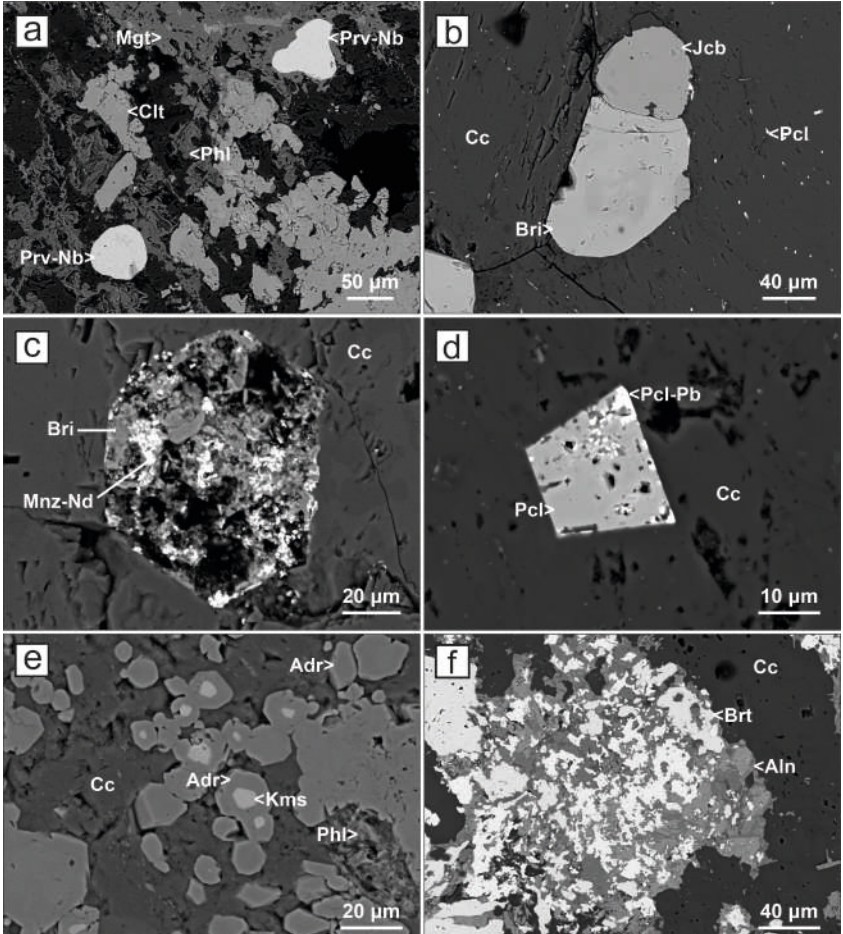


**Figure 3:** SEM (backscattered electron, BSE) images of the Fuerteventura carbonatites. **(a)** Subhedral
crystals of niobium-rich perovskite (Prv-Nb) associated with phlogopite (Phl) and magnetite (Mgt)
aggregates. The association has been affected by secondary hydrothermal processes, leading to the
formation of celestine (Clt). **(b)** Typical subhedral crystal of jacobsite (Jcb) associated with britholite (Bri).
Both crystals are hosted in magmatic calcite (Cc), with numerous disseminated microcrystals of pyrochlore
(Pcl). **(c)** Partially altered subhedral grain of britholite (Bri) hosted in magmatic calcite (Cc). The alteration
process led to the formation of secondary REE phosphates such as monazite-Nd (Mnz-Nd). **(d)** Euhedral
crystal of pyrochlore (Pcl) hosted in calcite (Cc). Brighter areas developed on the grain's borders correspond
to plumbopyrochlore (Pcl-Pb) zonation. **(e)** Typical mineral association related to small skarn like areas
associated with carbonatites. Subhedral zoned crystals of andradite (Adr), hosted in calcite (Cc) and
phlogopite (Phl), with a significant Zr zoning leading to kerimasite (Kms) cores. **(f)** Typical low-
metamorphic alteration developed on carbonatites composed of allanite (Aln) aggregates hosted in calcite
(Cc) and associated with secondary baryte (Brt). Abbreviations of mineral names in all the pictures follow
the criteria proposed by Warr (2021).
**Figure 4**

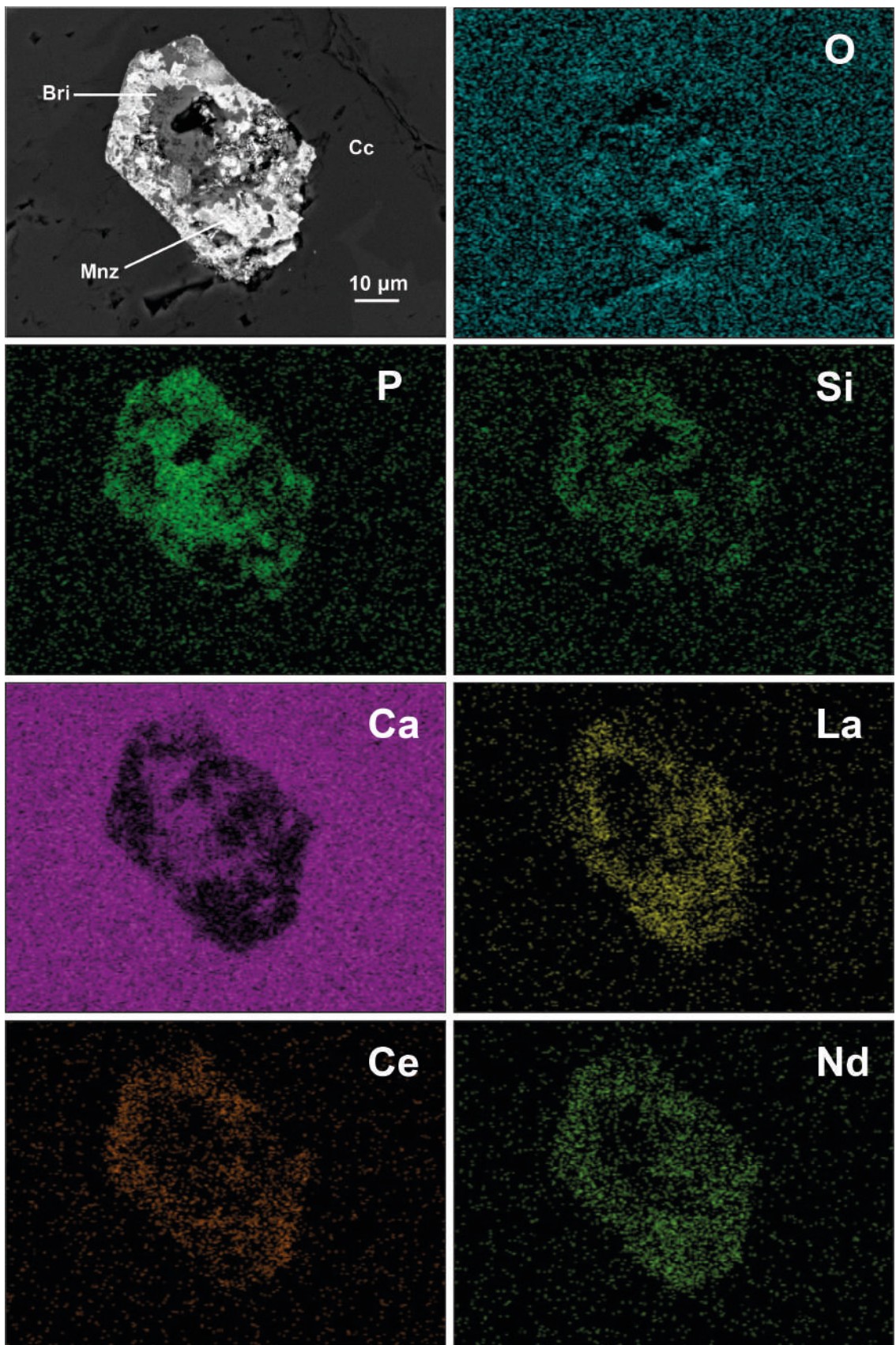


**Figure 4:** Wavelength-dispersive X-ray maps of representative compositional elements for an altered grain
of britholite (Bri) hosted in calcite (Cc) and partially transformed into secondary monazite (Mnz).
**Figure 5**

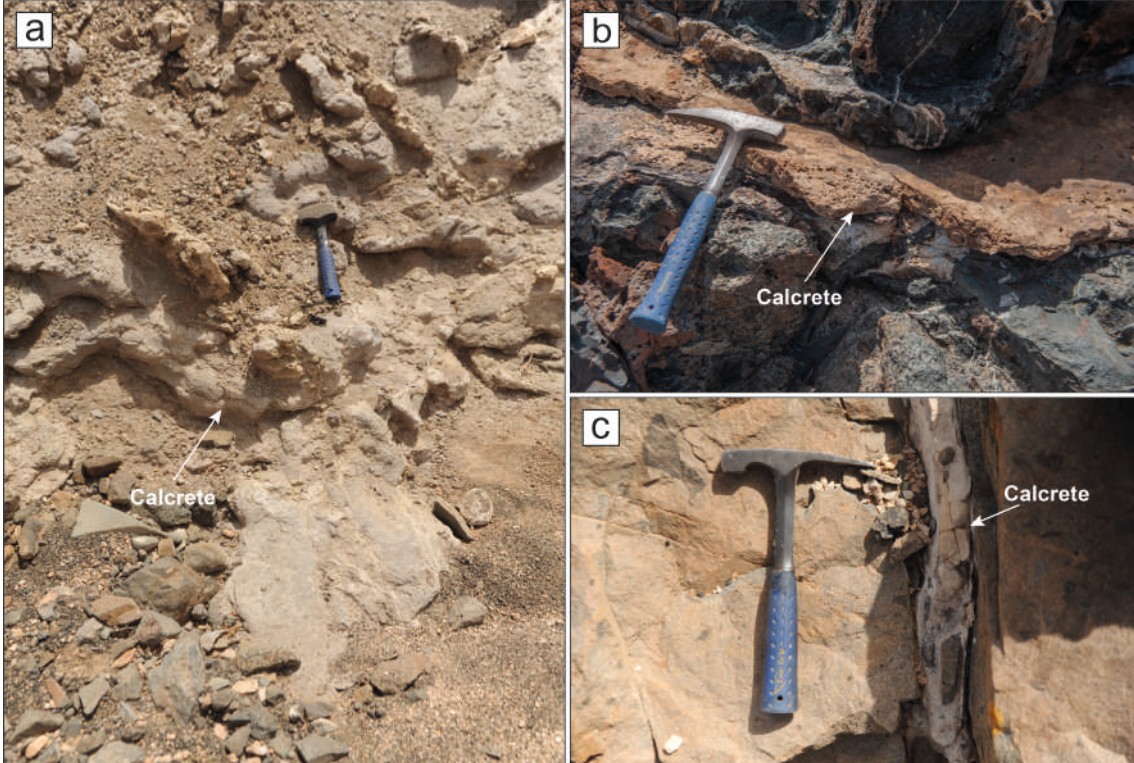


**Figure 5**: **(a)** General view of a typical surface outcrop of Quaternary calcrete located in the Aulagar ravine
area (profile B, sector 3; Fig.1). **(b)** Centimetre-thick calcrete layer filling a fracture between two
carbonatitic dikes in the Aulagar ravine area (profile B, sector 3; Fig.1). **(c)** Calcrete layer developed within
fractures between carbonatitic rocks in the Agua Salada ravine area (profile A, sector 1; Fig.1).












**Figure 6**

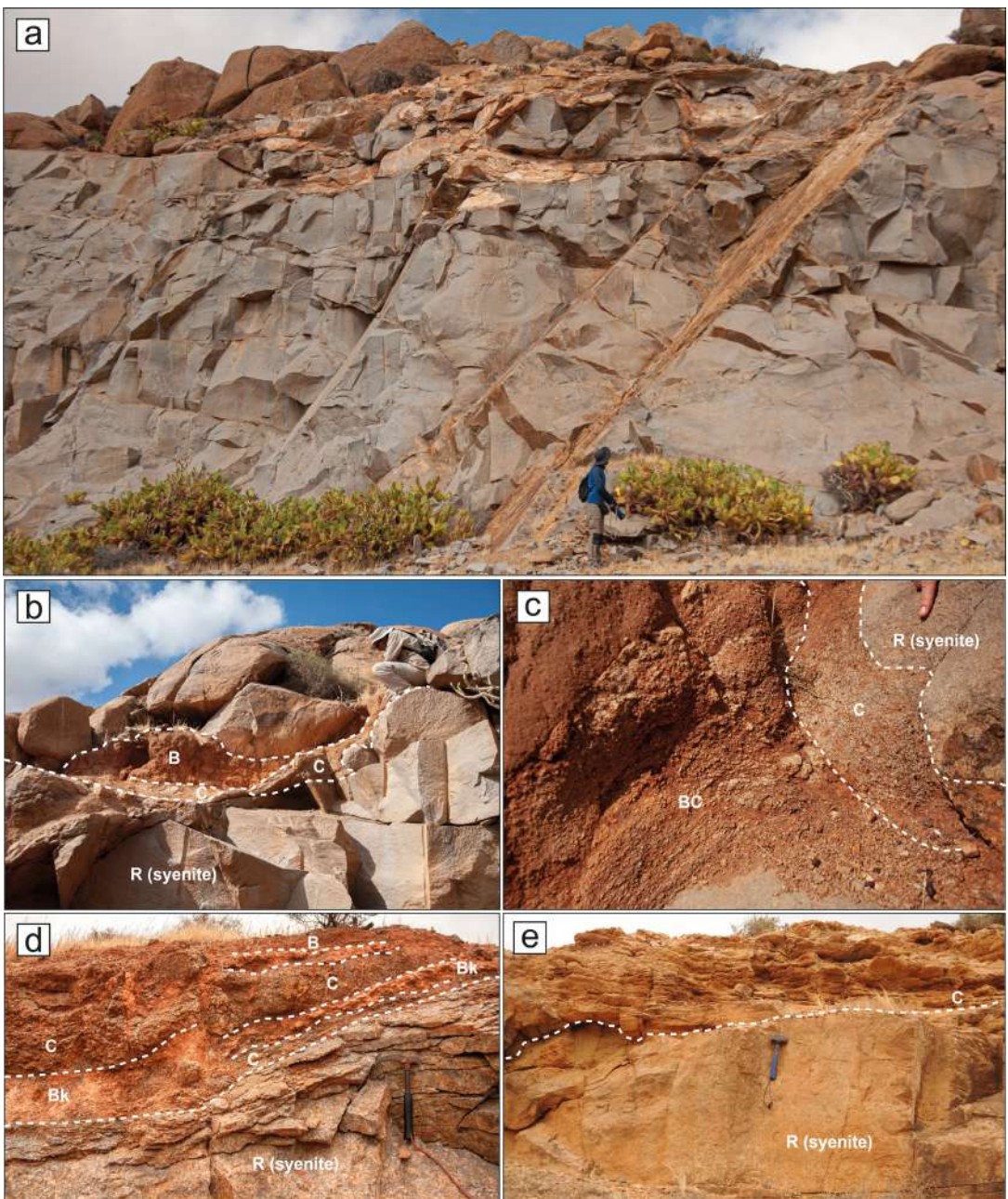


**Figure 6: (a)** General view of Las Peñitas quarry syenite outcrop (profile E, sector 2; Fig.1) where it is
possible to distinguish different fractures filled by injected secondary weathering products. **(b)** Syenite
weathering profile in Las Peñitas quarry (profile E, sector 2; Figure 1) showing surface erosion and B, BC
and C horizons injected in the syenite bedrock (R). **(c)** Weathering profile displaying the development of
C and BC horizons associated with a syenite protolith (R), located in Las Peñitas quarry (profile E, sector
2; Figure 1). **(d)** Weathering profile developed on syenite in the FV-30 road area (profile D, sector 2),
exhibiting the development of C, B and calcrete (Bk) horizons. **(e)** Weathering profile on syenite protolith
(R) displaying a metric sized C horizon in the Pájara area (profile F, sector 2; Figure 1).

**Figure 7**

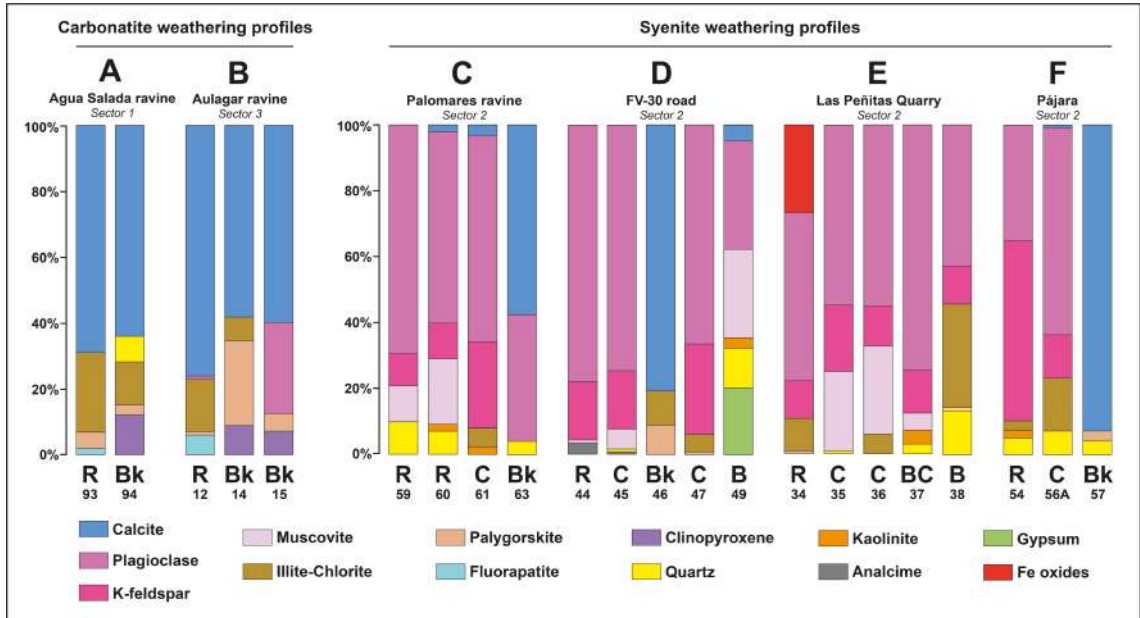

**Figure 7**: Graphical mineralogical quantification of the studied weathering profiles: (A) Agua Salada ravine; (B) Aulagar ravine; (C) Palomares ravine; (D) FV-30 road; (E) Las Peñitas quarry; (F) Pájara. The corresponding class assigned to the edaphic horizons (B, BC, Bk, C, R) and the sample number are shown at the foot of the columns.

**Figure 8**

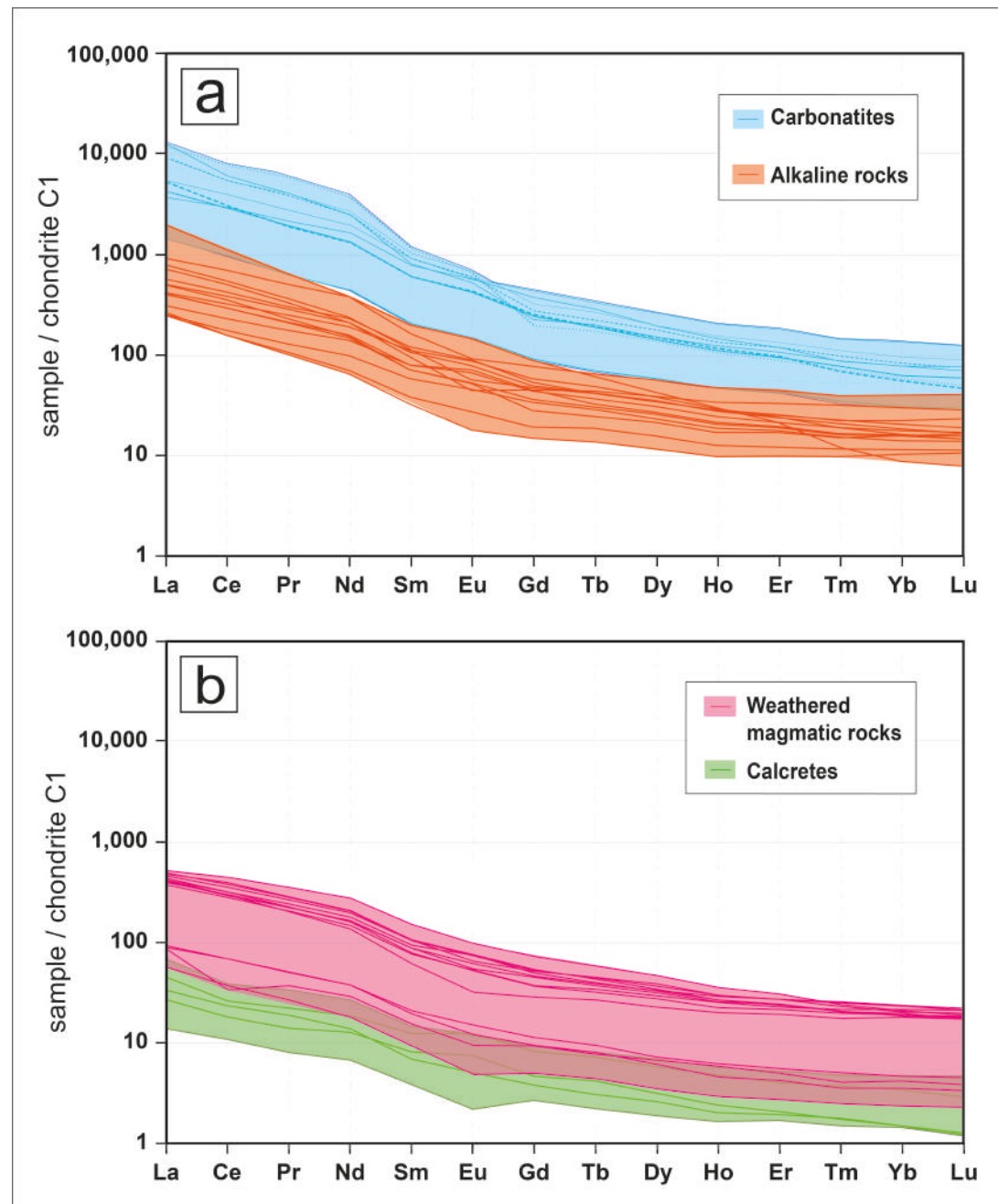



**Figure 8**: REE plots of the studied Fuerteventura lithologies normalised to C1 chondrites. Normalisation
values are from McDonough and Sun (1995).





**Figure 9**

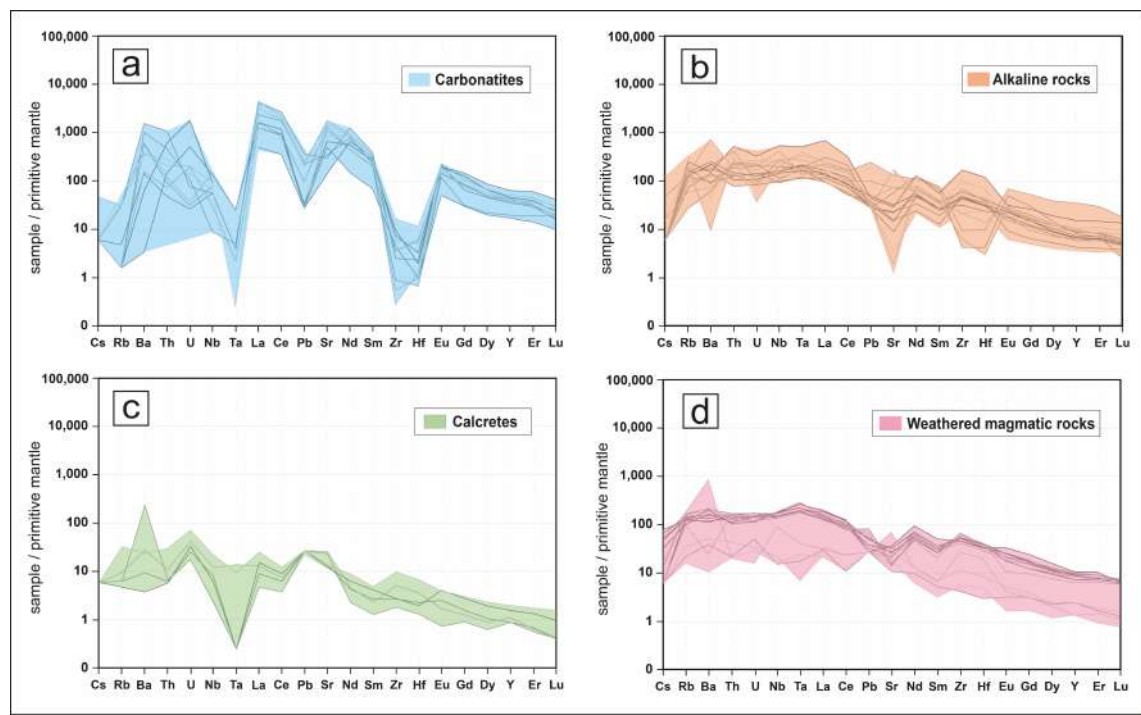

**Figure 9**: Multi-elemental trace element plots of Fuerteventura intrusive lithologies normalised to the primitive mantle. Normalisation values from McDonough and Sun (1995).

**Figure 10**

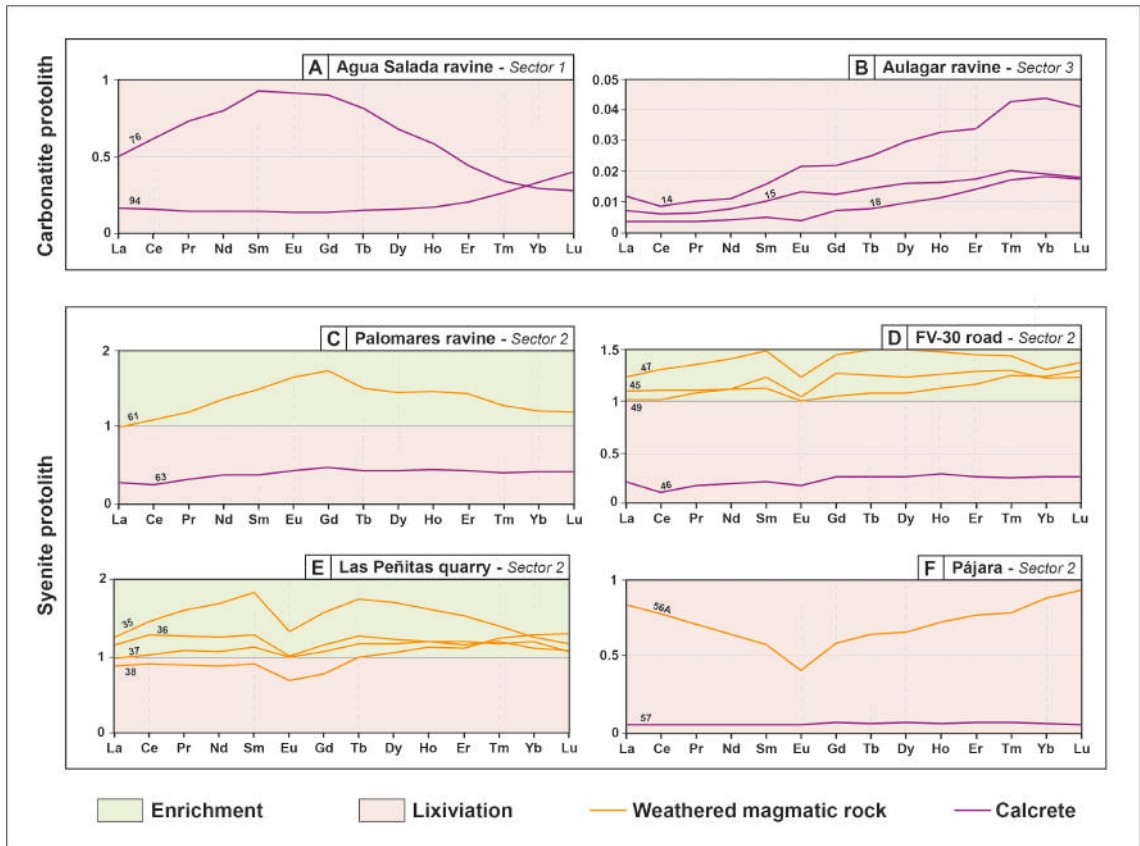

**Figure 10:** REE weathering enrichment/leaching diagrams between primary magmatic protoliths (carbonatites and syenites) and the associated weathering products from the studied profiles (Figure 1). The sample number is labelled on the corresponding pattern line.

**Figure 11**

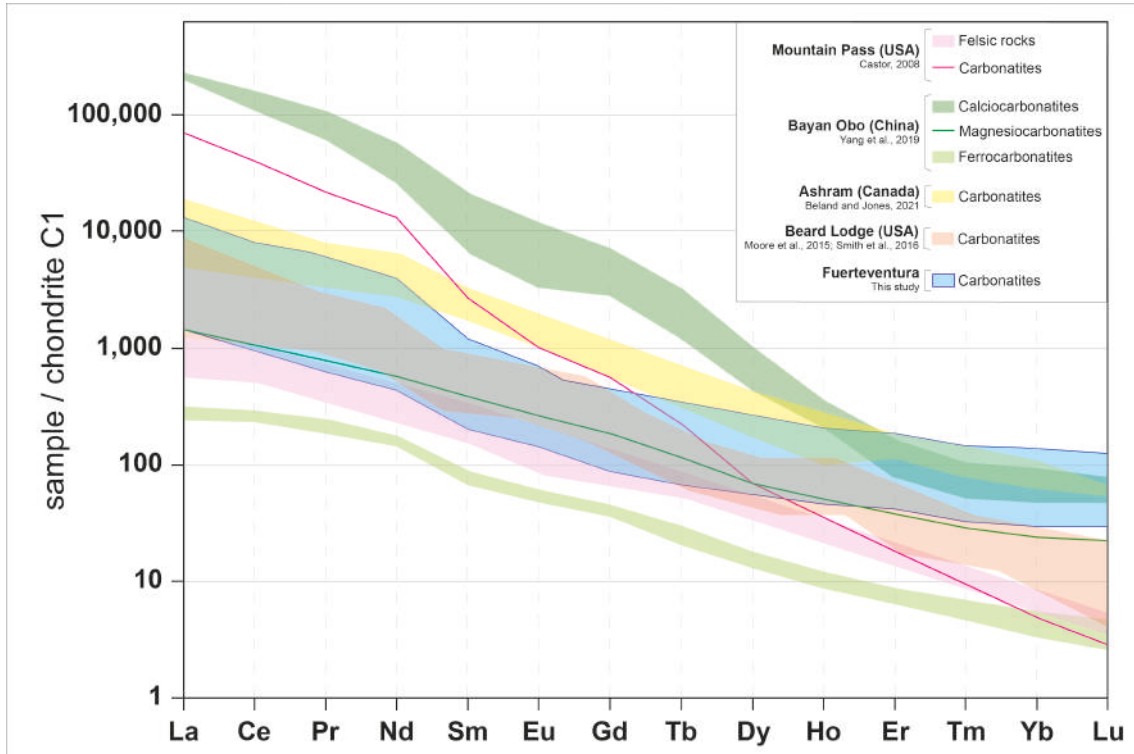


**Figure 11**: REE plot of the studied Fuerteventura carbonatites compared to other carbonatitic localities
worldwide where REE deposits have been reported. REE contents for comparison are from Castor (2008),
Yang et al. (2019), and Beland and Jones (2021). Normalisation values are from McDonough and Sun

1070 (1995).