# Peer review of "Rare earth element resources on Fuerteventura, Canary 1 Islands, Spain: a geochemical and mineralogical approach 2"

_EGUsphere, 2024_

## Author Comment (AC1)

Dear editor and authors,

The paper by Campeny et al is an interesting investigation of REE potential in the Fuerteventura carbonatites. This is also interesting because oceanic carbonatites have never been thoroughly assessed for REE mineralisation before (as far as I know). There is also scientific value in that the rocks obviously contain antiskarns, a very timely research topic (currently unrecognised by the authors, see comments below). I recommend the paper to be revised.

The authors greatly appreciate the overall positive comments on the article provided by Dr. Anenburg, as well as his corrections and proposed changes. We believe that they are very constructive and significantly enhance the entire manuscript. Now, we will proceed to address his specific observations, aiming to provide as detailed a response as possible.

**line 25:** Here, and after, please use significant figures correctly. A number like "10301.83" is meaningless because it implies that you know it to a precision of 0.01 ppm, and it's distinct to 10301.84 or 10301.32. Something like "about 10000 ppm" or even "about 1 wt%" should be ok, and please learn on correct usage of significant figures.

The authors agree with this comment and the entire manuscript concentration numbers have been amended according to it.

**line 28:** "enrichment" indicates a process. You just mean that the silicate rocks aren't as REE-rich as the carbonatites?

The authors agree with this comment. The term "enrichment" has been replaced by "contents".

**line 67:** Not accurate. Light REE are coming from magmatic rocks, primarily carbonatites. Heavy REE are coming from soils and weathering products.

The authors agree with this comment. The sentence has been amended according to this good appreciation.

**line 72:** The 50% limit by Le Maitre is restrictive and misleading. The carbonatite community is moving away to define carbonatites as rocks that form from carbonate melts, regardless of carbonate mineral content. See Yaxley et al 2022 here http://www.annualreviews.org/eprint/JVDPC4NDH4BAFJNWDAVJ/full/10.1146/annurev-earth-032320-104243

The authors agree with this comment. This sentence has been amended and we also added the proposed reference.

**line 73:** "basic" is outdated - please use mafic, ultramafic.

The authors agree with this comment. This sentence has been correspondingly amended.

**line 86:** Since you're specifically talking about REEs in carbonatites, the processes outlined by Anenburg et al 2021 are probably very relevant here https://doi.org/10.2138/gselements.17.5.327 and also Yaxley et al 2022 discusses this in detail.

The authors agree with this comment. The references have been added accordingly.

**line 87:** This paragraph will benefit from reference to Weidendorfer's work, for exmaple https://doi.org/10.1130/G39621.1 and https://doi.org/10.1007/s00410-016-1249-5

The authors agree with this comment. The reference has been added accordingly.

**line 92:** Which "aforementioned approach"? You're doing mineralogy and geochemistry, but you hardly talked about any mineralogical and geochemical approach above.

First paragraphs want to introduce the topic of Rare Earths and their significance from a general point of view. However, this introductory sentence ("aforementioned approach) has been removed to avoid misunderstanding.

**line 123:** "The FBC unit..." (other places in the places where you can safely switch word order and get rid of the "of the"...)

The authors agree with this comment. This sentence has been amended.

**line 131:** Partial fusion of what? The country rock?

The authors agree that this term could generate misunderstanding. Considering that the intention of this section is only to report a general geological setting, the sentence has been simplified.

**line 139:** Dikes of what? Carbonatites? Something else? Implied by your next sentence but worthwhile to clearly say this.

The authors agree with this comment. This sentence has been amended complementing the information about the composition of the FBC dykes.

**line 141:** Can you mark on the figure exactly where the fenites and carbonatites are?

Unfortunately, fenitization areas cannot be easily recognized in the presented pictures and this is why they are not marked in the referenced figure.

**line 147:** "Pyroxenite" cannot really be a magma type - it is a cumulate rock formed by crystallisation from a magma, with the magma migratin elsewhere. Alternatively, "pyroxenite" metasomatic zones around carbonatites are probably antiskarns - see for example https://doi.org/10.2475/03.2018.03 or https://doi.org/10.1016/j.chemgeo.2022.120888

The reviewer is likely correct, and these zones are probably anti-skarn formations. However, in the FBC they haven't been thoroughly studied from this perspective in any previous research. As this is an introductory section on the geology of Fuerteventura, we believe we should adhere to the data provided by other authors and refrain from making any conjectures. Nonetheless, we understand and agree with the comment regarding the pyroxenites, so the sentence has been simplified to avoid this error.

**line 157-184:** If your paper is on REE, and these units have no REE, why are they in the paper? Consider removing this.

This section aims to provide an introductory text to the geology of Fuerteventura, intending to familiarize the reader with the main geological units of the island and to aid in the interpretation and comprehension of the geological map presented in Figure 1. Therefore, although these units have not been assessed for their Rare Earth Element (REE) content, it is not problematic to discuss them in this introductory section of the manuscript. Hence, despite appreciating the reviewer's observation, the authors have chosen not to eliminate this section and to maintain it in its original form.

**line 191:** Remove commas from this sentence

The authors agree with this comment. This sentence has been amended.

**line 192:** Again, the Anenburg et al 2021 paper is probably relevant here

The authors agree with this comment. The reference has been added accordingly.

**line 196:** Not only that, sometimes weathering can make a deposit into an economic one, with the two best examples being Araxa and Mount Weld: https://doi.org/10.1093/petrology/egae007 and https://doi.org/10.1016/j.jsames.2023.104311

The authors agree with the reviewer that both proposed references were appropriate in this sentence. In addition, Chandler et al., 2024, as a new cite, has been also included in the reference section.

**line 252:** Check usage of word "basic"

The authors agree with this comment. This sentence has been amended and the term "basic" has been replaced by "mafic".

**line 256:** "primarily aegirine-augite and biotite" - the "primarily" indicates there are other mafic minerals. What are they?

The authors agree with this comment, this sentence has been amended to avoid misunderstanding.

**line 258:** Described where? Here? If yes, then rephrase without "described" because it's implied...

The authors agree with this correction and the sentence has been correspondingly amended.

**line 295:** What you're describing are precisely "antiskarns", a very hot topic of research these days. Concept first introduced by Anenburg & Mavrogenes (2018) which I referenced above, also see in depth discussion in Yaxley et al 2022. Vasyukova and Willy-Jones also talk about this (not sure if they use the name "antiskarn" though). Some other examples where similar textures and styles are observed: https://doi.org/10.1016/j.lithos.2023.107231 https://doi.org/10.1016/j.lithos.2023.107480 https://doi.org/10.1016/j.lithos.2022.106647. The britholites you're seeing are also very typical. Experimentally recreated here: https://doi.org/10.1126/sciadv.abb6570 (see experiment CbSi), and also observed in nature here http://hdl.handle.net/1885/154263 There's also a paper should be out in Contributions to Mineralogy and Petrology on the topic very soon with thermodynamic modelling. If it's not out by the time you do the revision, then contact me and I'll send it to you.

The authors totally agree with this comment. Indeed, we agree that this term can be used to describe some areas related with the Fuerteventura carbonatites. Therefore, we have added a small paragraph commenting on it and included some of the references suggested by the reviewer.

**line 300:** Correct spelling is sulfates, not sulphates (f- spelling endorsed by the UK Royal Society of Chemistry for example). Also IMA-approved spelling is baryte, not barite

The authors appreciate this comment. Both terms have been amended in the entire manuscript.

**line 320:** Reduction in illite? This means you some to begin with, but this is the first time you're mentioning illite and chlorite

The authors agree with the reviewer's comment. There was a mistake in the sentence and it has been correspondingly amended.

**line 329:** Why don't you just say which elements, instead of referring to a supp table?

The authors agree with this comment. The sentence has been amended to avoid misunderstanding. However, we consider that the reference to Table S4 is useful and it has not been removed.

**line 333:** Please use this for typical crustal values https://doi.org/10.1016/B978-0-08-095975-7.00301-6

The authors accept this comment. Reference values for comparison have been replaced and used from Rudnick and Gao, 2014.

**line 337:** See previous comment on significant figures. You can leave this precision for the supp tables (as long as uncertainty is reported with it), but not in the main text.

The authors agree with this comment and the entire manuscript concentration numbers have been amended according to it.

**line 352:** You still haven't said which elements these are. What is "significant"?

The authors agree with the reviewer that the sentence is not very accurate and it does not provide significant information. Then, we decided to remove it to avoid misunderstanding,

**line 355:** How do you reconcile it with the fact that you had pyrochlore? Could this be an analytical artifact?

The authors agree with the reviewer that the results of HFSE are underestimated. For sure it is an analytical artifact generated by the poor digestion of pyrochlore. The section has been amended clarifying this point.

**line 387,395:** Significant figures

The authors agree with this comment and the entire manuscript concentration numbers have been amended according to it.

**line 388:** Use Rudnick, not Balaram

The authors accept this comment. Reference values for comparison have been replaced and used from Rudnick and Gao, 2014.

**line 422-427:** I recommend removing this. The topic is much more complex than you put it, and not really in the scope of your manuscript.

The authors agree with this comment. This sentence and their references have been correspondingly removed.

**line 434:** Why is this noteworthy? Calcite is overwhelmingly the most common mineral in all carbonatites worldwide. This is akin to saying that cpx is noteworthy in basalt or quartz in granite...

The authors completely agree with the reviewer that the sentence was not well written. This has been amended according to this comment.

**line 503:** This cannot be understated - Fuerteventura is an UNESCO biosphere reserve!
**line 538:** One final sentence could be useful here: "Given the non-exceptional REE grade of Fuerteventura compared to other deposits, most REE being hosted in unexploitable and refractory britholite, irregularly distributed mineralisation with low overall tonnage, and Fuerteventura being a UNESCO biosphere reserve, we conclude that economic development of any REE resources on the island is extremely unlikely to occur." And something similar in the abstract as well. Just saying, because on a bigger scale you don't want to start any unnecessary hype that Fuerteventura is "the next big thing" because no good can come out of this.

We had already considered the need to clarify this aspect as a final culmination of the discussion. In fact,

we already included a paragraph in the original manuscript (see lines 497 to 504) about this topic. However, we have adapted our final discussion sentences as well as the abstract, with some phrases proposed by Dr. Anenburg that we believe enhance the initial writing and provide greater solidity to the arguments presented.

---

## Author Comment (AC2)

The authors present a summary of the geology of Fuerteventura and present mineralogical and geochemical data to evaluate the viability of some of the rocks and their weathered counterparts as potential sources of the REE. The paper is an interesting thought exercise, but based on the results, the authors should be absolutely clear that these rocks have 0% chance of ever being a REE mine – even on a small scale.

The authors appreciate the reviewer's positive feedback and are pleased that our study has been found interesting. However, we would like to clarify that our work is part of a scientific research, focused on the detailed characterization of the mineralogy and geochemistry of these particularly exotic rocks, within a geological context that has received scant attention. Our study does not, in any capacity, aim to conduct an economic assessment of these lithologies for the purposes of a mining project. This task falls outside the scope and objectives of our research. Therefore, we believe that such a comment is not applicable.

The authors focus on the carbonatites as these have the highest grade – reaching 1 wt.% total rare earths in a single grab sample. While some comparison is made to REE grades in existing REE mines, such a comparison is disingenuous, and in many deposits 1% REE would barely make the cut-off grade.

The authors believe that the use of the term "disingenuous" is completely inappropriate. Our comparative analysis is not disingenuous, nor has it been approached from the perspective of economic geology. Our goal was to juxtapose the concentrations found in the richest sample from Fuerteventura against those from other well-documented locations with significant concentration data. It is clear that a comprehensive economic geology study and the viability assessment of these lithologies as ore deposits would require a broader range of factors to be considered. However, we wish to reiterate, that is not the purpose of our article. Our focus is not on evaluating the mining potential of Fuerteventura's rocks; instead, we present a mineralogical and geochemical comparison with other locations of international renown.

Moreover, the samples presented are relatively mineralogically complex, with the REE spread across three different mineral – with little indication of which mineral would be a focus. The carbonatite bodies are small and discontinuous, further rendering them an economic non-starter. Lastly, all the rocks studied appear to be located in a protected area – perhaps outside of the scope of the thought exercise, but an important point nonetheless, and one that should probably be made and emphasized in case a lay-reader might misinterpret the paper.

The authors regard our study as a presentation of unpublished data in the fields of mineralogy and geochemistry. We reiterate that we are not conducting a mining study. As we have already emphasized in the discussion, conclusions, and abstract of the paper, any mining assessment would require much more information and must adhere, as it can only be, to the current environmental and land management regulations and constraints. We believe these points are sufficiently clear, do not lead to misunderstandings, and are well-argued. Moreover, they have been complemented with the constructive comments of Dr. Anenburg (Review 1 on egusphere). We consider that they do not require more further clarification.

I was surprised by the lack of attention given to the weathered alkaline rocks. Considering these have a profile of a metre thick in places, could there be thicker weathering profiles elsewhere? Should the authors wish to take the thought exercise further, they may way to look into some of the samples as potential ion-adsorption type deposits. Interestingly, only a few of the samples actually seem to have weathered to clay, so I suspect that there will not be a large amount of easily leachable REE, but it may still be worth an inquiry.

The authors disagree with the reviewer's comment. The alteration profiles of alkaline rocks have been described and studied from a mineralogical and geochemical perspective, as can be seen in the results and discussion sections of the manuscript, as well as in the corresponding figures (see, for example, Figures 6 and 7). The study options proposed by the reviewer, which could be very valid, would once again be focused on mining studies. We would like to emphasize again that mining evaluation is not the objective of this article.

Lines 38-53, could be condensed to a few sentences.

The authors believe that this introduction is suitable. It assists the reader in understanding the general context about REEs and offers relevant information that highlights the research objectives. Introductory information can be always condensed (or expanded) based on authors preferences. In this case, we do not see the need to summarize it.

Line 76, the example minerals you give here are fluorcarbonates.

The authors agree with the reviewer and the sentence has been amended according to this comment.

Line 115: remove 'ago'

The authors agree with the reviewer and the sentence has been amended according to this comment.

Line 126: 'associated to' à 'associated with'

The sentence has been amended according to this comment.

Line 319: 'carbonatite profiles'… on line 308-311, you mention that there is no weathering profile associated with the carbonatites, except for the development of calcrete veins. Perhaps you should be more specific on line 319, and say that the samples are of calcrete. How does the formation of these calcrete relate/differ from the calcretes mentioned on L180-184, described as forming from calcarenites, rather than an igneous precursor?

The authors appreciate these comments from the reviewer as they exemplify that certain aspects regarding the relationship between calcretes and rocks of the FBC were not entirely clear. Calcretes are spatially associated with rocks from the Fuerteventura Basal Complex (carbonatites, syenites, etc.), but not genetically. What has been assessed in this work is whether these spatially associated materials have had chemical interaction, especially concerning REEs. To clarify this point, the manuscript has been modified in accordance with the reviewer's insightful comments and questions.

Line 407: It is disingenuous to take the average value of 2581 ppm REE across the whole complex, when the areas which actually define the resource are much higher concentration. No-one would take the felsic rocks from around the mountain pass area into a resource calculation, and the carbonatites have LREE

contents over an order of magnitude higher than the Fuerteventura samples. A single sample of 1 wt% REE, while high for the Canary Islands, would barely make the cut-off grade for many carbonatite-hosted REE deposits. It is also somewhat disingenuous to compare grab samples (especially the highest-grade grab samples) and compare these with resources from a select handful of other carbonatites. The values from most other carbonatites will reflect average grades over an area considered economically feasible to mine.

The authors strongly disagree with the reviewer's assessments. We never claimed that Fuerteventura's carbonatites are economically comparable to other deposits. Our aim is simply to conduct a geochemical comparison of our samples with those from other carbonatites globally, where more extensive data is available. While we acknowledge that our sampling is limited compared to these deposits, we believe our comparisons are illustrative, transparent, and contribute to understanding REE resources in oceanic carbonatites like those in Fuerteventura. We always cite our data sources and present our contribution modestly, aiming for a comprehensive and honest global perspective. While other comparisons or approaches might be more suitable, we reject the notion that our work, based on objective data comparison, is disingenuous.

Line 414: check full stop after 'Figure 11)'

The authors agree with the reviewer and the sentence has been amended according to this comment.

Line 419-427: I wouldn't make too much of this relatively flat HREE profile. The HREE are challenging to extract from carbonatites, and where these profiles are elevated, are commonly hosted in a different mineral to those which can be exploited commercially, and consequently lost during minerals processing. See https://doi.org/10.1016/j.mineng.2020.106617

Line 428-433: That the REE are split between three discreet phases only means they will be diluted further during processing.

The authors consider that these two considerations are made from the perspective of mining treatment and the economic benefit of the mineralizations. As we have reiterated in different answers, this is not the objective of the current article. Our aim has been to characterize the geochemistry and mineralogy of these lithologies in relation to REEs, but we have not conducted any mining study based on the potential economic exploitation of the mineralizations.

Line 434-441: I don't quite follow the logic here. What relevance does the presence of calcite have on the presence of REE-fluorcarbonates?

The authors agree with the reviewer that the sentence was no clear. In fact, the first sentence about calcite was removed according to the comments of Dr. Anenburg, and now has been amended clarifying that REE carbonates are, indeed, fluorcarbonates.

Lines 462-465: the examples given are all of carbonatites with significant weathering and the development of regolith up to (and over) 100 m thick. The carbonatites in Fuerteventura have developed calcrete veins up to a few cm, locally. Why make the comparison?

The authors consider this comparison to be valid, as it has been made to provide the reader with information through examples where there is enrichment in REEs in different lithologies associated with alteration processes, not necessarily in carbonatites. In fact, in the case of Las Mercedes, for instance, this enrichment is associated with karstic bauxites.

Lines 443-472: There's no consideration given here to ion-adsobtion type deposits which, given only the weathering profile above the syenite is developed more than a few cm, could perhaps be of consideration. Ion adsorption deposits require at least 50% of the REE in the weathering profile to be easily leachable using a medium pH reagent, such as ammonium sulfide. In these cases, the REE are loosely bound to clays developed on the weathering profile, and can be easily stripped from the clay and recovered. Ion adsorption type deposits have much lower cut-off grades where relatively cheap in-situ leaching can be applied, and low-grade resources can be economic – especially where HREE contents are high. I am surprised that this avenue hasn't been explored.

We appreciate the consideration provided by the reviewer and will take it into account for future research on these lithologies. However, in the present article, the extraction of REEs through clay treatment at the plant is not an objective of our research. Although it is a very interesting topic, it is beyond the scope of our current investigation.

Lines 475-477: Based on the geochemical data, maybe, but based on the field observations, it is clear that the extremely small size of these bodies does not warrant any further investigation.

Once again, the reviewer confuses basic research studies on geochemistry, mineralogy, distribution of critical elements, lithologies, etc., with studies on mining and economic exploitation of mineralizations. We reiterate that our research is not focused from this perspective, as has already been emphasized in different sections of the manuscript and in this response document.

Line 477-478: Grade is not everything. Size and mineralogy are important too. A large, mineralogically amenable, low grade deposit can be much better than a small, mineralogically complex, high grade body.

The authors agree with this comment. However, these criteria are important from the perspective of mineral treatment and the beneficiation of rare earths in the mining industry. Our work is not focused from this perspective, and this is why these aspects have not been addressed.

---

## Author Response (AR2)

**Topic editor decision: Publish subject to minor revisions (review by editor)** by Johan Lissenberg

A number of comments from Reviewer 2 relate to the feasibility (or otherwise) of the Fuerteventura carbonatites as an economic and exploitable source of REE. In response, the authors write that 'our work is part of a scientific research, focused on the detailed characterization of the mineralogy and geochemistry of these particularly exotic rocks, within a geological context that has received scant attention. Our study does not, in any capacity, aim to conduct an economic assessment of these lithologies for the purposes of a mining project.'

However, as is, the revised manuscript still contains a clear economic geology rationale, and is not entirely framed around understanding REE in ocean island carbonatites, as suggested by the response. For instance, the abstract frames the context as the EU 'actively promoting exploration of REE resources', and states that the paper comprises a 'preliminary evaluation as potential targets for REE exploration'. Similarly, the Introduction talks about the how EU Critical Raw Materials Act 'aim to establish a comprehensive framework to ensure a secure and sustainable supply of CRMs, including REEs, in the coming years'. The title also reflects this ('Rare earth element resources on Fuerteventura').

I would encourage the authors to update the manuscript, particularly the abstract and introduction, clarifying the framework in which they see their work. If the paper is about the Fuerteventura carbonatites as (potentia) REE resources then some of reviewer 2's comments should be addressed; if it's about understanding REE in ocean island carbonatites then the abstract and introduction should reflect that. Please edit the manuscript accordingly.

Dear editor,

We kindly appreciate your comments and considerations. Then, we have attempted to address them through a series of significant modifications, particularly applied to the title, abstract, introduction, and conclusions, as well as minor adjustments in the results and discussion sections. We believe these changes contribute to greater coherence throughout the manuscript and align with the feedback provided by the anonymous reviewer as you suggested.

We hope this revision meets your expectations, and you find our work suitable for publication in Solid Earth.

Sincerely,

Marc Campeny

---

## Author Response (AR3)

13 May 2024
**Executive editor decision: Publish subject to technical corrections**

Dear Dr. Campeny, I thank you for the integration of all remarks and suggestions arisen during the three rounds of revision. I'm glad to accept your paper for publication, following the advice provided by the Dr. Lissenberg, Topical Editor, after very minor technical corrections. Thank you again for submitting the results of your research to Solid Earth.

Best regards,

Andrea Di Muro, SE Executive Editor

Correction 1: a very very minor technical modification: at line 463 I suppose is the oxygen (18) and not the carbon isotope; please modify this minor misprint

Correction 2: at the beginning and along the whole paper, a more detailed description of the lithological composition of calcrete can be useful, in order better to clarify their genetic links with the other lithologies

Dear editor,

We greatly appreciate your overall comments and celebrate the acceptance of this article in Solid Earth.

It has been a pleasure to go through this review process. We sincerely believe that all the changes and suggestions made have contributed to enhancing our initial work.

Finally, we have also incorporated some additional final changes in response to the two specific corrections mentioned in your latest message, which we hope will be favorably considered.

Sincerely,

Marc Campeny